# Separable pupillary signatures of perception and action during perceptual multistability

**Jan W Brascamp[1,2]\*, Gilles de Hollander[3], Michael D Wertheimer[1], Ashley N DePew[1], Tomas Knapen[4,5]**

[1]Michigan State University, Department of Psychology, East Lansing, United States; [2]Michigan State University, Neuroscience Program, East Lansing, United States; [3]Zurich Center for Neuroeconomics, Department of Economics, University of Zurich, Zurich, Switzerland; [4]Vrije Universiteit Amsterdam, Amsterdam, Netherlands; [5]Spinoza Centre for Neuroimaging, Royal Academy of Sciences, Amsterdam, Netherlands

**Abstract** The pupil provides a rich, non-invasive measure of the neural bases of perception and cognition and has been of particular value in uncovering the role of arousal-linked neuromodulation, which alters both cortical processing and pupil size. But pupil size is subject to a multitude of influences, which complicates unique interpretation. We measured pupils of observers experiencing perceptual multistability—an ever-changing subjective percept in the face of unchanging but inconclusive sensory input. In separate conditions, the endogenously generated perceptual changes were either task-relevant or not, allowing a separation between perception-related and task-related pupil signals. Perceptual changes were marked by a complex pupil response that could be decomposed into two components: a dilation tied to task execution and plausibly indicative of an arousal-linked noradrenaline surge, and an overlapping constriction tied to the perceptual transient and plausibly a marker of altered visual cortical representation. Constriction, but not dilation, amplitude systematically depended on the time interval between perceptual changes, possibly providing an overt index of neural adaptation. These results show that the pupil provides a simultaneous reading on interacting but dissociable neural processes during perceptual multistability, and suggest that arousal-linked neuromodulator release shapes action but not perception in these circumstances.

**\*For correspondence:**
brascamp@msu.edu

**Competing interests:** The authors declare that no competing interests exist.

## Introduction

The brainstem's neuromodulatory systems can profoundly influence cognitive functions by altering neural response properties within the cortical circuits that mediate those functions (*Aston-Jones and Cohen, 2005*; *Sara and Bouret, 2012*; *Lee and Dan, 2012*; *Pfeffer et al., 2018*). Recent work has used multistable visual stimuli to examine neuromodulatory influences on visual processing. Such stimuli cause perception to alternate between different interpretations of the sensory data, and recent work suggests that the noradrenergic arousal system associated with the brainstem's locus coeruleus impacts perception in this situation, perhaps by altering the response gain of visual cortical neurons involved (*Einhäuser et al., 2008*; *Sara and Bouret, 2012*; *Laeng et al., 2012*; *Kloosterman et al., 2015a*; *Pfeffer et al., 2018*). We used a new combination of experimental methods to evaluate this idea.

Like previous researchers, we focused on pupil size changes that accompany switches between alternative percepts, because pupil dilations can non-invasively convey noradrenaline release (*Murphy et al., 2014*; *Joshi et al., 2016*; *de Gee et al., 2017*). Existing results reveal a transient

pupil dilation accompanying perceptual switches, suggestive of a noradrenaline surge (*Einhäuser et al., 2008*; *Hupé et al., 2009*; *Kloosterman et al., 2015b*; *de Hollander et al., 2018*). The characteristics of this dilation, including its relation to the temporal dynamics of the perceptual sequence, have formed the basis for theorizing on the role of the locus coeruleus and associated structures in perceptual multistability, and on the role of arousal in perception more broadly (*Einhäuser et al., 2008*; *Hupé et al., 2009*; *Sara and Bouret, 2012*; *Laeng et al., 2012*; *Kloosterman et al., 2015b*; *de Hollander et al., 2018*). But what complicates interpretation of the published results is that perceptual switches always involved multiple neural events spaced closely in time—some related to perception and some not—which makes it difficult to tie pupillary measures back to any specific event (see *Hupé et al., 2009* for a similar assessment). In particular, perceptual switches in existing work were always task relevant—observers overtly reported them or, in some cases, covertly tracked them—so that each switch included both the perceptual change and further task-related processing. Of note, noradrenaline-related pupil dilations have been linked to numerous cognitive factors that may be at play in such a situation: motor planning, attentional reorienting, altered cognitive load, and surprise, among others (*Kahneman and Beatty, 1966*; *Aston-Jones and Cohen, 2005*; *Hupé et al., 2009*; *Laeng et al., 2012*; *de Gee et al., 2014*; *Wang and Munoz, 2015*). As such, it is unclear how published switch-related pupil signals map onto specific perceptual and cognitive processing steps, and it is unclear whether any part of those signals is tied to mechanisms that shape perception, rather than to processes that underlie task execution generally.

In light of the above we evaluated switch-related pupil signals in a set of conditions that included conditions where switches were irrelevant to the observer. We used binocular rivalry, a form of multistability in which perception alternates between two interpretations that each correspond to a stimulus shown to only one of the two eyes (*Blake and Logothetis, 2002*). To isolate and quantify distinct components that might be reflected in pupil size in association with perceptual switches, we employed four conditions in a two-by-two factorial design (*Figure 1A*). The first factor was the nature of the perceptual changes: they could either be endogenously generated in response to binocularly incompatible input (*Rivalry* conditions; 'LE' and 'RE' are left and right eye, respectively), or be exogenously prompted via on-screen 'replay' animations designed to resemble the binocular rivalry experience (*On-screen* conditions). The second factor was task-relevance: observers were asked either to manually report perceptual changes when they happened (*Report* conditions) or to instead perform a task to which the perceptual changes were irrelevant (*Ignore* conditions; the task was a peri-threshold detection task involving small transients in both eyes' displays simultaneously – events whose timing was uncorrelated with that of the perceptual switches, and which did not affect pupil size in a way that impacts our conclusions; *Appendix 1—figure 1*). We reasoned that comparisons between the *Rivalry* and *On-screen* conditions would help tease apart signals linked to the mechanism of endogenous perceptual switches and signals linked to perceptual changes generally (an idea copied from numerous functional imaging studies; for example, *Lumer et al., 1998*), whereas comparisons between the *Ignore* and *Report* conditions would help distinguish signals related to perceptual changes from signals related to factors such as reorienting, surprise, and report.

To preview our main results, we found both rivalry switches and on-screen switches to be accompanied by a similar pupil response, but we found this response to be markedly different between the *Report* and *Ignore* conditions. The *Ignore* response consisted of a constriction tied to the perceptual change itself (even though no net change in light flux was involved), whereas the *Report* response was composed of this perception-related constriction component as well as an overlapping dilation component linked to the behavioral report. Whereas the report-related dilation is plausibly associated with transient noradrenaline release, the perception-related constriction is not. We further found that these two response components differ in their relationship to the timing of the perceptual sequence. Specifically, it is the perception-related constriction, rather than the report-related dilation, that shows a robust dependence on this timing. These results indicate that pupil signals during perceptual multistability include two overlapping but separable components: both a dilation that is plausibly related to arousal-linked noradrenaline release at a physiological level and to task execution at a behavioral level, and a constriction that accompanies visual cortical processes closely tied to perception.

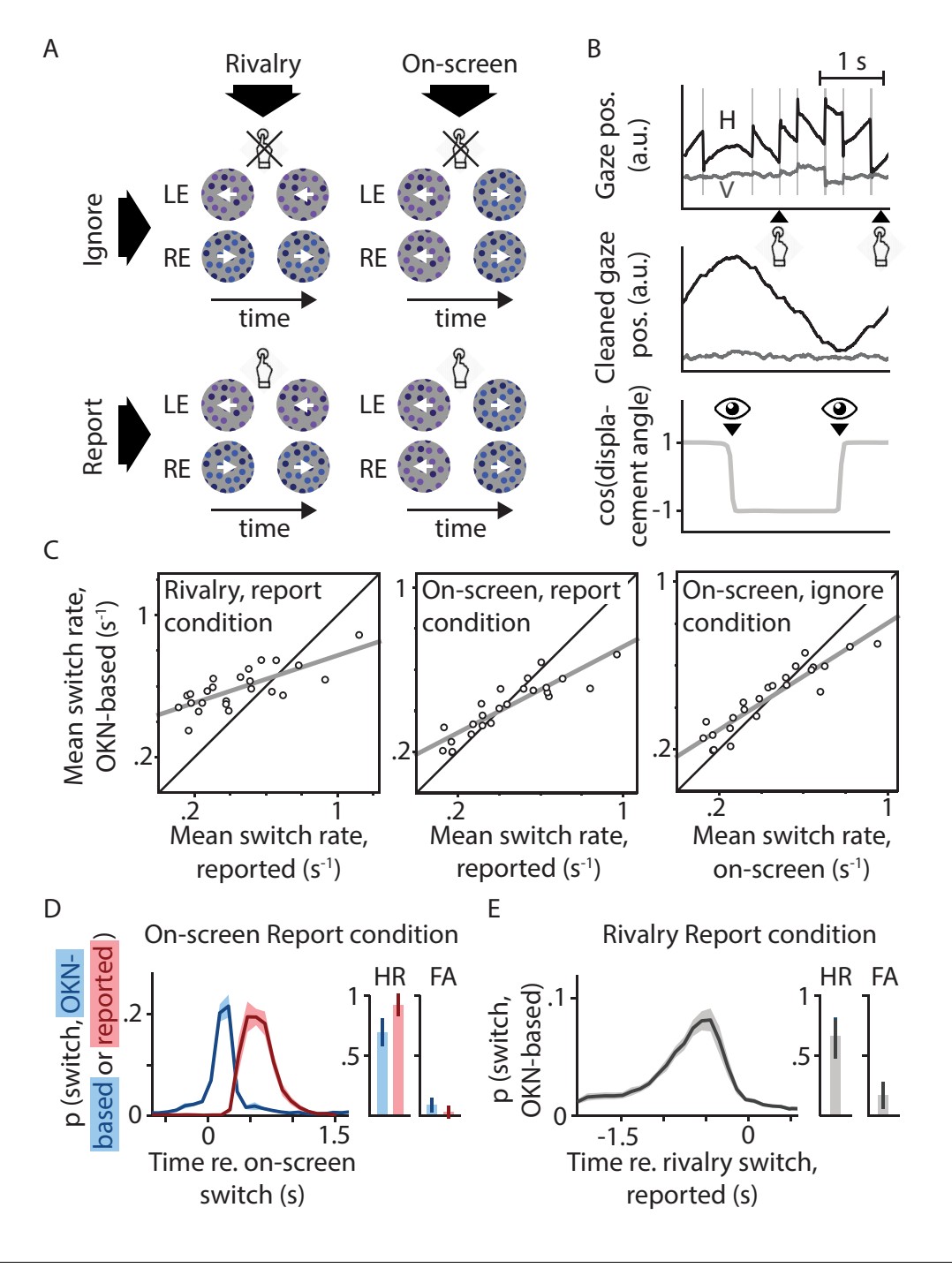

**Figure 1.** Experimental design and behavioral validation. (**A**) Our two-by-two factorial design included both binocular rivalry and on-screen replay, and both conditions where switches in perception were behaviorally relevant and ones in which they were not. (**B**) Perceptual switch moments in all conditions were identified based on reflexive eye movements (OKN) in response to the moving stimulus. (**C**) The per-observer numbers of switches identified using this method correlated strongly with the numbers of switches estimated based on manual report or on-screen switch events. Pearson's *r* values from left to right: 0.75, 0.89, and 0.92; all p<0.0001. (**D and E**) There was a tight correspondence between the timing of perceptual switches as estimated using these three methods, such that the area under each curve within the 1.5 s period with the bulk of the probability mass (hit rate; HR) was several times higher than the area under each curve across the surrounding 1.5 s (false alarm rate; FA; see Materials and methods for details).

# Results

Twenty-four observers were included in our analyses. We relied on involuntary eye movements to identify perceptual switch events, even in the absence of manual report. To this end the visual stimulus area was filled with dots that could translate either leftward or rightward. In the *Rivalry* conditions the two eyes' dots moved in opposite directions; in the *On-screen* conditions motion direction physically alternated in the visual display (*Figure 1A*). Previous work has shown the direction of reflexive eye movements in response to visual motion (optokinetic nystagmus, or OKN) to form a reliable indicator of perceived motion direction in similar situations, not just for binocularly congruent motion but also during binocular rivalry (*Fox et al., 1975*; *Leopold et al., 2001*; *Naber et al., 2011*; *Frässle et al., 2014*; *Aleshin et al., 2019*). We verified that this was also the case here (*Figure 1B–E*). Per-observer estimates of perceptual switch rate were highly correlated (see figure caption) although not identical (see Discussion) between our OKN-based measure and measures based on either manual report or, in the *On-screen* conditions, replayed direction reversals (*Figure 1C*). Moreover, there was a tight temporal correspondence between the moments of perceptual switching as identified by the three different methods (*Figure 1D–E*), providing further confidence in our OKN method's suitability. Opposite dot motions were also associated with different dot colors (but not different dot luminances) to promote perceptual exclusivity during binocular rivalry, that is to counteract perceptual mixtures of both eyes' displays (*Knapen et al., 2007*). We verified that our findings on switch-related pupil responses did not importantly depend on which color, or which eye, became dominant during the switch (*Appendix 1—figure 2*).

We used a general linear model approach to deconvolve pupil responses associated with perceptual switches in each condition. An important benefit of such an approach over, for instance, an 'event-related averaging' approach that involves averaging pupil signals across time windows anchored to events of interest, is that general linear model approaches are suitable for isolating the response associated with a given event type, even in situations where, in practice, that response frequently overlaps with pupil responses tied to nearby events (e.g. *Dale, 1999*; *Wierda et al., 2012*). This is helpful in this case because several pupil-linked events (switches, blinks, key presses) may occur within the time it takes the pupil response associated with one such event to unfold (several seconds). To facilitate between-condition comparison, we centered our analyses on switch events as identified using our OKN measure—the only measure available in all four conditions. While, in general, changes in gaze direction can be associated with changes in pupil size, both real and apparent when using video-based eye trackers (*Gagl et al., 2011*; *Wang and Munoz, 2015*; *Knapen et al., 2016*; *Laeng and Alnaes, 2019*), control analyses rule out the possibility that our observations are importantly related to the association between perceptual switches and ocular events in our paradigm (*Appendix 1—figure 3*). To address the possible concern that our results may be impacted in a relevant way by limitations in the OKN-based switch detection algorithm, we furthermore compared results across different switch indices (OKN-based, manually reported, and on-screen) within the specific conditions that allow such comparisons, and found that pupil response patterns were highly similar irrespective of the switch index used (*Appendix 1—figure 5*).

*Figure 2A* shows pupil area as a function of time during the time period surrounding perceptual switch moments (inferred from OKN), averaged across observers (for the corresponding per-observer data, please see *Appendix 1—figure 6*). Visual inspection of the curves suggests a qualitative difference between the *Ignore* conditions (top row of plots) and the *Report* conditions (bottom row of plots). For both of the *Ignore* conditions perceptual switches are accompanied by a rapid drop in pupil size (marked in *Figure 2A* as 'C1', for 'constriction 1'), followed by a rapid recovery back to near baseline ('D1', for 'dilation 1'. Note that the terms 'constriction' and 'dilation', as used here to mark specific parts of the pupil response, refer to periods during which pupil size decreases and periods during which pupil size increases, respectively. In this context, the terms do not specify whether the net pupil size is smaller or larger than baseline during those periods). For both *Report* conditions, on the other hand, visual inspection suggests a more complex pupil response. Although the initial constriction/re-dilation sequence (marked in the plots) is visible, the amplitude of the initial constriction is small enough for the re-dilation to go well past baseline, leading to a final, more gradual constriction back to near baseline later ('C2', for 'constriction 2'). Such added complexity would be consistent with the fact that the *Report* conditions include an additional event, the key press report, on top of the perceptual change that these conditions share with the *Ignore* conditions.

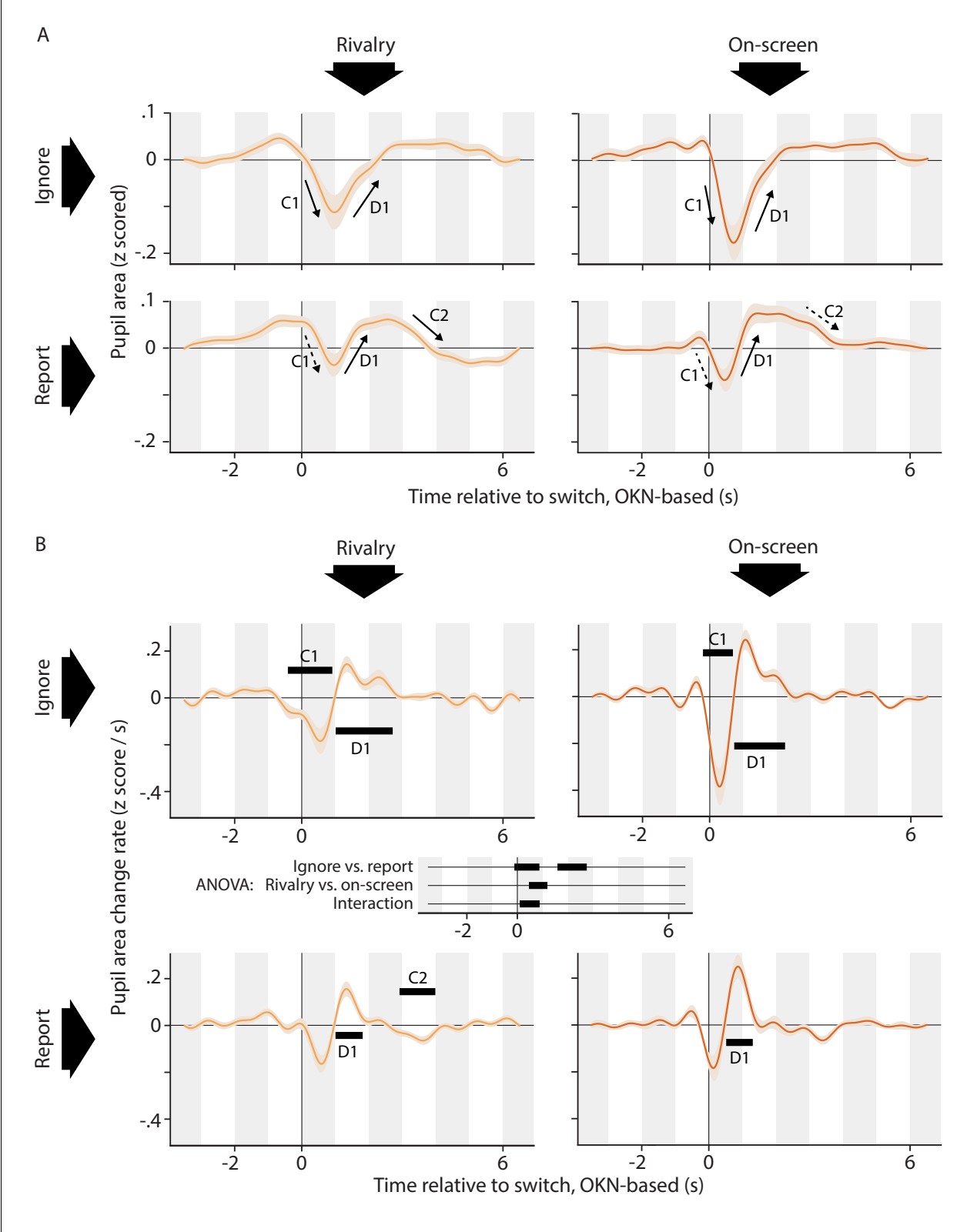

**Figure 2.** Pupil responses time locked to the perceptual switch. (**A**) Pupil area as a function of time around the moment of the perceptual switch for all four conditions. (**B**) Rate of pupil size change on the same time axis. Black bars within each plot of panel B show time periods during which this rate differs from zero (cluster-level p<0.01). At the center of panel B, between the plots, is a time axis that shows the results of a repeated-measures ANOVA comparing the four conditions, with black bars indicating significant differences (cluster-level p<0.01). All conditions are associated with an initial

*Figure 2 continued on next page*

*Figure 2 continued*

constriction (C1; significant only in the *Ignore* conditions) and subsequent dilation (D1; significant in all conditions). Only in the *Report* conditions does that dilation go substantially pass baseline, to be followed by a second constriction back to baseline (C2; significant in one condition). Consistent with the idea of a more dilation-dominated response in the *Report* conditions, the pupil change rates reach a lower negative extreme in the *Ignore* conditions as compared to the *Report* conditions, yet not as a high a positive extreme. Responses are qualitatively similar between the *Rivalry* and *On-screen* conditions, but more pronounced in the latter conditions. This is again borne out by examining the extremes: change rates in the *On-screen* conditions reach both a lower minimum and a higher maximum. Negative extremes: −0.18 z-score/s (*Rivalry Ignore*), −0.16 z-score/s (*Rivalry Report*), −0.38 z-score/s (*On-screen Ignore*), −0.18 z-score /s (*On-screen Report*). Positive extremes: 0.14 z-score/s (*Rivalry Ignore*), 0.16 z-score/s (*Rivalry Report*), 0.24 z-score/s (*On-screen Ignore*), 0.25 z-score /s (*On-screen Report*). All confidence intervals, both in this plot and elsewhere in the paper, show across-observer standard errors of the sample mean.

*Figure 2B* shows the rate of pupil area change on the same time axis as used in *Figure 2A*. Statistics were performed on the rate of change rather than on pupil size itself; a choice that follows previous work (*Joshi et al., 2016*; *de Gee et al., 2020*) and that is consistent with the notion that pupillary change (rather than size) is a more immediate marker of transient neural events, which alter the input to the antagonistic iris musculature rather than altering pupil size in a step-wise fashion (*Reimer et al., 2016*). Black bars within each plot of panel B indicate periods of significant change over time within individual conditions (based on one-sample t-tests). In addition, the time axis at the center of panel B, between the plots, denotes periods during which the rate of pupil change significantly differs between conditions (based on an ANOVA). Throughout the paper all effects that are marked as significant have a cluster-based p<0.01 (see Materials and methods for details).

The data summarized in *Figure 2B* support the qualitative impressions described in relation to *Figure 2A*, and provide statistical context. The ANOVA results (*Figure 2B*; center) indicate that the data from the *Ignore* conditions differ significantly from those from the *Report* conditions throughout most of the first three seconds following the switch event, overlapping with both the initial constriction and the subsequent dilation. The figure panels themselves show that the initial constriction (C1) reaches significance in both *Ignore* conditions, while being visible only as a non-significant period of negative size change in the *Report* conditions. This is consistent with the above-mentioned impression that this constriction is smaller in the *Report* conditions. The subsequent dilation (D1), on the other hand, is significant in all conditions. A later period of negative size change (C2) is visible between the 2 s mark and 4 s mark for both *Report* conditions, and reaches significance in the *On-screen Report* condition.

The ANOVA results also indicate a, less extensive, difference in pupil response between the *Rivalry* conditions and the *On-screen* conditions immediately following the switch (overlapping both with the end of the initial constriction and the start of the redilation), as well as an interaction in a slightly earlier time period (overlapping mainly with the initial constriction). A comparison between the curves of *Figure 2A* and between the curves of *Figure 2B* (left vs. right columns of plots) shows no clear qualitative difference between these two groups of conditions but, instead, suggests that on-screen switches and rivalry switches are both associated with a response of the same general shape, but that the *On-screen* response is more rapid and pronounced. One possible explanation for this is that switches in the *On-screen* conditions may be more abrupt, which would likely make for a closer temporal alignment of switch-related pupil responses across switches, resulting in a more articulated, less smeared out, estimated pupil response.

*Figure 3* gives an impression of the degree of consistency, at the level of individual observers, of the across-observer effects marked in *Figure 2B*. The two-by-two layout of *Figure 2B* is repeated in *Figure 3*, and *Figure 3* summarizes per-observer data for each time period (C1, D1, C2) during which *Figure 2B* displays the pupil area change rate as significantly different from 0 (i.e. for each time period marked with a black bar along the curves of *Figure 2B*). In *Figure 3* each dark gray circle shows the average value of an individual observer's pupil area change rate within such a time period (labeled at the top). Each light gray horizontal line, in turn, shows the median of per-observer averages. There is some degree of circularity in this analysis: because the time windows were selected on the basis of these same data (assessed at the across-observer level) it stands to reason that the general tendency in the per-observer data matches that of the across-observer average. Nevertheless, the clustering of the per-observer data points (gray circles) and the positioning of the median value of those data points (gray horizontal line) show that the significant effects marked in

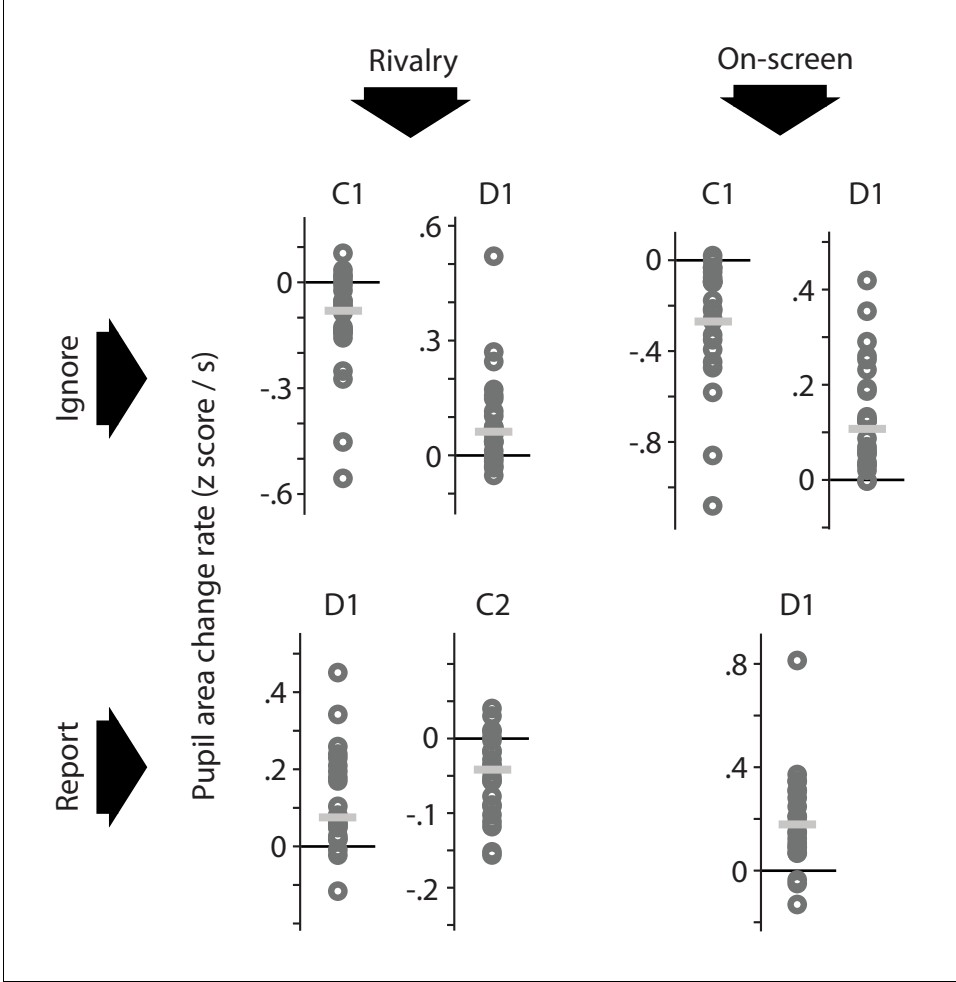

**Figure 3.** Per-observer summary data corresponding to the effects marked as significant (based on across-observer statistics) in the plots of *Figure 2B*. For each time period during which the pupil area change rate differs significantly from 0, as marked in *Figure 2B*, this figure shows the average pupil area change rate of each individual observer (dark gray circles) as well as the median of those per-observer averages (light gray horizontal lines). For each of the time intervals, the majority of individual observers numerically show an effect in the same direction as the across-observer average, which indicates that across-observer significance here arises from data patterns that are fairly consistent across observers rather than being carried by a select number of outliers.

the plots of *Figure 2B* are not carried by a select number of outliers, but rather are numerically present in a majority of individual observers. In particular, for each time period examined in *Figure 3*, both the majority of per-observer data points and (equivalently) the median of those data points lie on the same side of 0 as the corresponding across-observer average shown in *Figure 2B*. In other words, most individual observers numerically show a constriction/dilation whenever the across-observer average shows a significant constriction/dilation. For full per-observer pupil response curves, rather than their average levels within select time periods, please see *Appendix 1—figure 6*.

The finding that all conditions show a similar rapid constriction (C1) and re-dilation (D1) suggests that this component of the pupil response is tied to the occurrence of a (spontaneous or replayed) perceptual switch: the common factor across all conditions. This suggestion is strengthened by the close temporal correspondence between the onset of initial constriction and the moment at which perception switches (the temporal reference used in *Figure 2*, that is the OKN-based estimate of the switch moment, typically falls within half a second of the perceptual change; see *Figure 1D*). The fact that only the *Report* conditions show the re-dilation going well past baseline and being followed by a final constriction (significantly so for the *Rivalry Report* condition), suggests an additional, and later, dilation associated with manually reporting the perceptual switch. This dilation, although being

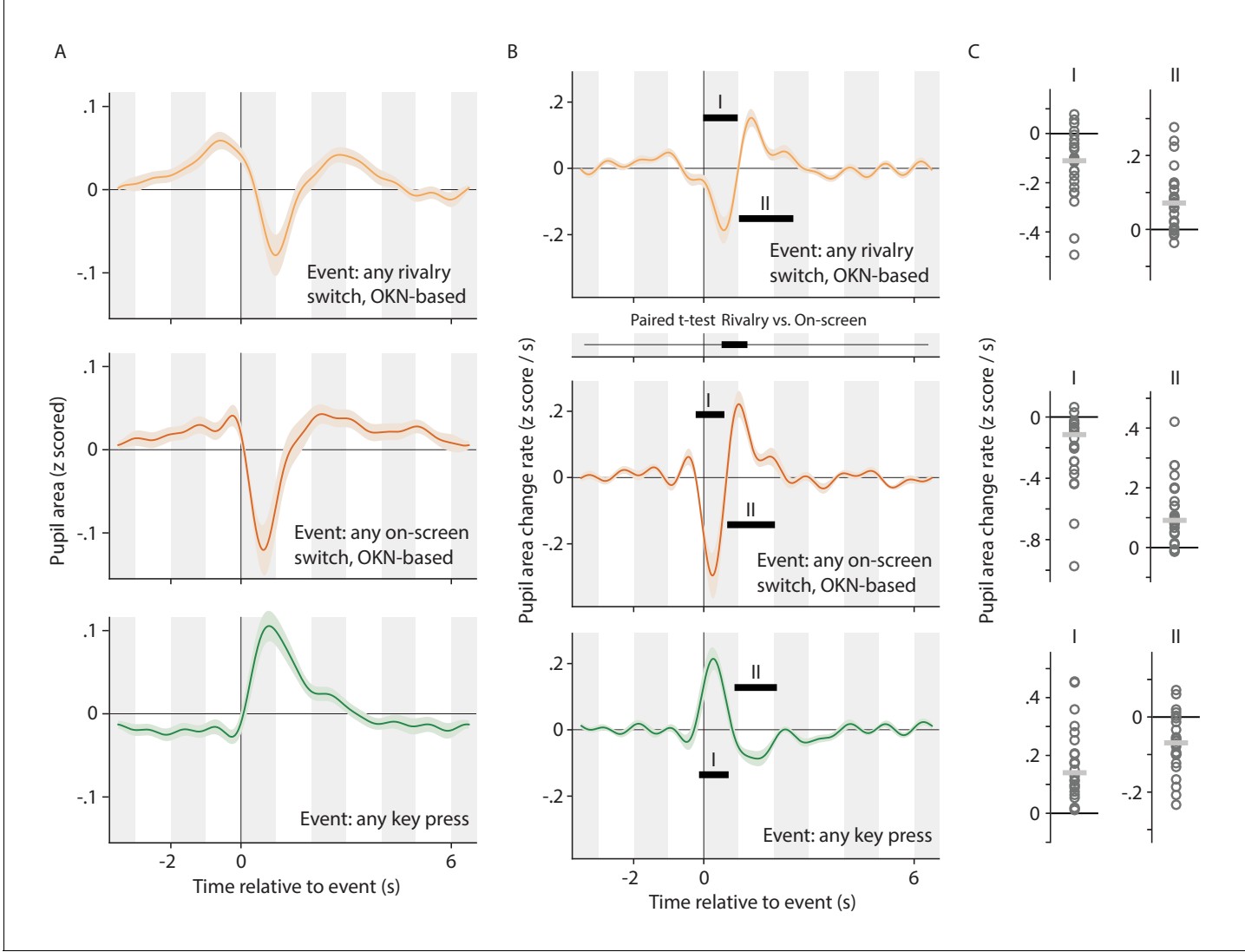

**Figure 4.** Alternative analysis of the data underlying *Figure 2*, now concatenating pupil signals across conditions, and regressing key press events and (rivalry or on-screen) switch events separately. This allows the pupil response associated with switches to be teased apart from the pupil response associated with key presses, even in the data from the *Report* conditions where the two consistently overlap. All plotting conventions in panels A and B are as in *Figure 2*, but the time axis between the top and center plot of column B now shows the results of a paired t-test. All plotting conventions in panel C are as in *Figure 3*.

delayed relative to the constriction, would nevertheless overlap with the constriction and dampen it, explaining the net smaller (and non-significant) initial constriction amplitude observed in *Figure 2* in association with the *Report* conditions, as compared to the *Ignore* conditions. These observations are consistent with the idea of a dilation component associated with the key press report, given the fact that manual report typically follows both the perceptual event and its OKN-based concomitant by an interval that ranges between hundreds of milliseconds up to about a second (*Figure 1D*), and therefore often happens while the perception-related constriction is still ongoing.

To more closely examine the shapes of the individual pupil response components that appear to contribute to the overall patterns shown in *Figure 2*, we next analyzed the data in a way designed to tease apart the putative switch-related pupil constriction and report-related pupil dilation. For this analysis, we concatenated, for each observer, all pupil data across all four conditions into a single time course, and deconvolved three pupil responses: one associated with *Rivalry* switches irrespective of whether they were reported or not (i.e. including switches across both the *Report* condition and the *Ignore* condition in the same regressor), one associated with *On-screen* switches

irrespective of whether they were reported or not, and one associated with key presses irrespective of whether they reported a perceptual switch or something else. In particular, during the *Ignore* conditions key presses were not in response to perceptual switches, but to subtle visual changes in the peri-threshold detection task that observers performed in those conditions (see Materials and methods). For this analysis, key presses of that latter type were included in the same regressor as key presses in the *Report* conditions. In other words, this analysis treated the pupil time course as the combined sum of both switch-related and key-related response components, and their temporal independence in the *Ignore* conditions allowed those components to be resolved separately in spite of their temporal association in the *Report* conditions.

The results of this analysis are shown in *Figure 4*, using the same format as *Figures 2* and *3*. Both rivalry switches (top plot in panels A and B) and on-screen switches (center plot in panels A and B) are marked by a rapid constriction and re-dilation immediately following the switch. The response to on-screen switches is, again, more rapid and pronounced, leading to a significant difference between the responses during a brief time window immediately following the switch (panel B; time axis between the top and center plot). Key presses, on the other hand, are accompanied by a qualitatively different (and also significantly different; not shown) pupil response, characterized by a rapid dilation and then re-constriction back to near baseline (bottom plot in each panel). These results are consistent with the interpretation, articulated above, that the biphasic pupil responses (constriction, then dilation, and then return to baseline) observed in our *Report* conditions (*Figure 2*, bottom row in each panel) reflect a superimposition of these two separate components. *Figure 4C* shows per-observer data corresponding to each of the significant effects marked along the curves of *Figure 4B*, analogous to what *Figure 3* showed in relation to *Figure 2B*. This again reveals a considerable level of consistency at the level of individual observers, with a large majority of individual observers numerically showing a dilation/constriction whenever a significant dilation/constriction is marked in *Figure 4B*.

Authors of previous studies have observed a biphasic pupil response for reported switches that is qualitatively similar to the one we found (*Einhäuser et al., 2008*; *Naber et al., 2011*; *de Hollander et al., 2018*; see also *Einhäuser, 2016*). Still, published interpretations and analyses have tended to focus on switch-related *dilation* (probably because the dilation is generally more pronounced, with some studies reporting no constriction at all; *Hupé et al., 2009*; *Kloosterman et al., 2015b*). This tendency is intertwined with the literature's emphasis, discussed above, on interpretations in terms of noradrenaline release from brainstem arousal systems, which would lead to dilation. A further factor tied into noradrenaline-centered interpretations is an observed relation between switch-related pupil responses and the temporal dynamics of the perceptual time course. Specifically, using only conditions where perceptual switches were manually reported or otherwise task-relevant (see Discussion), previous work has shown the net amplitude of pupil dilation following a given perceptual switch to vary with the duration of the immediately preceding perceptual dominance episode (*Kloosterman et al., 2015b*; *de Hollander et al., 2018*), as well as with the duration of the immediately following one (*Einhäuser et al., 2008*; *de Hollander et al., 2018*, although see *Hupé et al., 2008*). The former finding fits well with the fact that noradrenaline release in response to a given event depends on the degree of predictability of the event, given that an earlier end to a dominance episode is less predictable than a later one (*Kloosterman et al., 2015b*; *de Hollander et al., 2018*; see Discussion). The latter finding, in turn, is consistent with the fact that noradrenaline release leads to increased neural gain and altered circuit dynamics in the cortex (*Aston-Jones and Cohen, 2005*; *Gilzenrat et al., 2010*; *Sara and Bouret, 2012*), which could influence the emergence of further perceptual switches (*Einhäuser et al., 2008*; *Sara and Bouret, 2012*; *Laeng et al., 2012*; *Kloosterman et al., 2015a*; *Pfeffer et al., 2018*). In other words, existing work has primarily treated switch-related pupil signals as noradrenaline-related dilations, and has aimed to fit observed relationships with perceptual dynamics into this framework. However, existing work has not attempted to separate individual components of the switch-related pupil signal. Our above results suggest that the net dilation observed in that work may well correspond to a superimposition of both a dilation and a constriction which are shifted in time by the observer's reaction time for each report. Based on the available data, therefore, it is unclear whether the observed association with the temporal dynamics of the perceptual cycle stems from the dilation component, which would support an account in terms of noradrenaline release, or whether it stems from the constriction component,

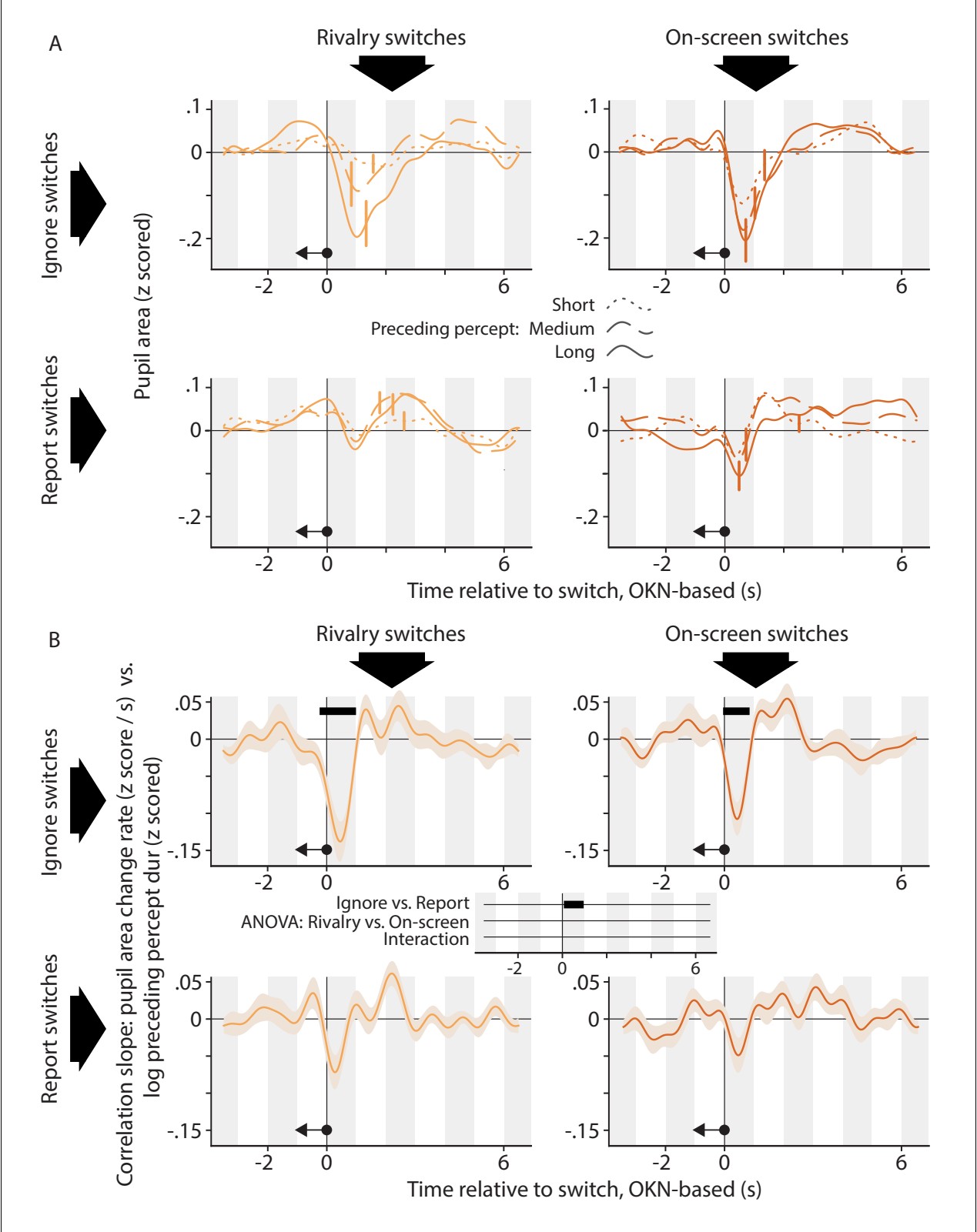

**Figure 5.** The relation between the pupil response associated with a perceptual switch and the duration of the perceptual dominance duration that preceded that switch. (**A**) Each plot shows, for a different condition, pupil size surrounding the moment of a perceptual switch, separated out into three equal-sized groups of perceptual switches on the basis of the preceding dominance duration (dotted curves: short; dashed curves: medium; solid curves: long). (**B**) Each plot shows, for a different condition, how the rate of pupil size change correlates, per time point in the interval that surrounds

*Figure 5 continued on next page*

*Figure 5 continued*

perceptual switches, with the duration of the preceding dominance duration. We performed inferential statistics only on the data of panel B; not A. All plotting conventions are as in *Figures 2* and *4*.

which would not be consistent with any existing account and would call for interpretation. Our next analyses were designed to address this question.

*Figures 5* and *6* focus on pupil responses separately per condition. For both A panels, we grouped switch events into three separate regressors, corresponding to three quantiles of the immediately preceding (*Figure 5A*; symbolized by the left-pointing arrow in each plot) or subsequent (*Figure 6B*; symbolized by the right-pointing arrow in each plot) dominance duration, from brief to long. In the former case (*Figure 5A*) this analysis suggests a more pronounced switch-related pupil constriction for switches that terminate a longer dominance episode, especially in the *Ignore* conditions (top row). To quantify and statistically evaluate this impression, *Figure 5B* shows how the rate of pupil size change varies with preceding dominance duration. Here, we did not separate switch events into quantiles but instead included as a covariate the (normalized) durations of the perceptual dominance episodes immediately preceding the switch events. In other words, we investigated whether, at any time point within our deconvolution window, between-switch variability in the preceding dominance duration was correlated with between-switch variability in pupil area change rate. Qualitatively all four plots show a dip shortly after the switch, consistent with a stronger constriction following longer dominance episodes, but this dip is significantly deeper in the *Ignore* conditions (time axis at the center between the four plots), and only reaches significance in those conditions, for both *Rivalry* and *On-screen* switches.

The analyses underlying *Figure 6* are the same as those underlying *Figure 5*, but center on the perceptual dominance episode that follows, rather than precedes, a perceptual switch. Here, panel A shows only modest differences in pupil area between the three groups of perceptual switch events, now separated into quantiles of the subsequent perceptual dominance duration rather than the preceding one. Panel B is consistent with this observation. It provides some evidence that subsequent dominance duration is positively correlated with pupil area change rate around 2 s following the perceptual switch in the *Rivalry Ignore* condition, but statistical support for this is not quite compelling. Specifically, the effect is significant when assessed within that condition (top left plot), but the size of the effect does not differ between conditions (time axis at the center between the four plots; no effects reach our chosen alpha level of p<0.01, and the time period in which the significant within-condition effect is observed is associated with a *Rivalry vs. On-screen* difference with p=0.30 and a still weaker interaction). This lack of any between-condition differences precludes strong conclusions, both because in general terms the relevance of significant within-condition effects is predicated on the presence of between-condition differences, and because in this specific case no meaningful effect can exist in the *On-screen* conditions (because brain processes cannot influence the timing of on-screen events). Because authors of previous studies (*Einhäuser et al., 2008*; *de Hollander et al., 2018*) who reported correlations with subsequent dominance duration during rivalry used somewhat different methods, we had a closer look at our data using methods more similar to theirs. First, the existing work involved statistics on pupil size rather than on its temporal derivative, so we repeated the ANOVA of *Figure 6B* using the correlation with pupil area as the dependent variable, but we again found no between-condition differences (the smallest cluster-level p-value, p=0.40, occurred around 2 s before the switch in the *Active vs. Passive* comparison). Second, because existing work used key presses as the marker of perceptual switch timing, we repeated the t-test of the *Rivalry Report* condition with key presses, rather than OKN, as the index of perceptual switches (for the *Rivalry Ignore* condition key presses are not available). The correlation between pupil size change rate and subsequent duration showed no significant differences from 0, although the smallest p-value (p=0.16) did occur for a cluster of positive correlations around 2 s after the reported switch; a data pattern reminiscent of that shown for the *Rivalry Ignore* condition in *Figure 6B*. In sum, although our data do not strongly argue against a relation between the pupil response and subsequent dominance duration, they provide no convincing evidence in favor, either.

To summarize, our analyses provide evidence that switch-related pupil constriction, which occurs in isolation in the *Ignore* conditions, depends on the duration of the preceding perceptual dominance episode. Evidence that report-related dilation depends on preceding dominance duration, or

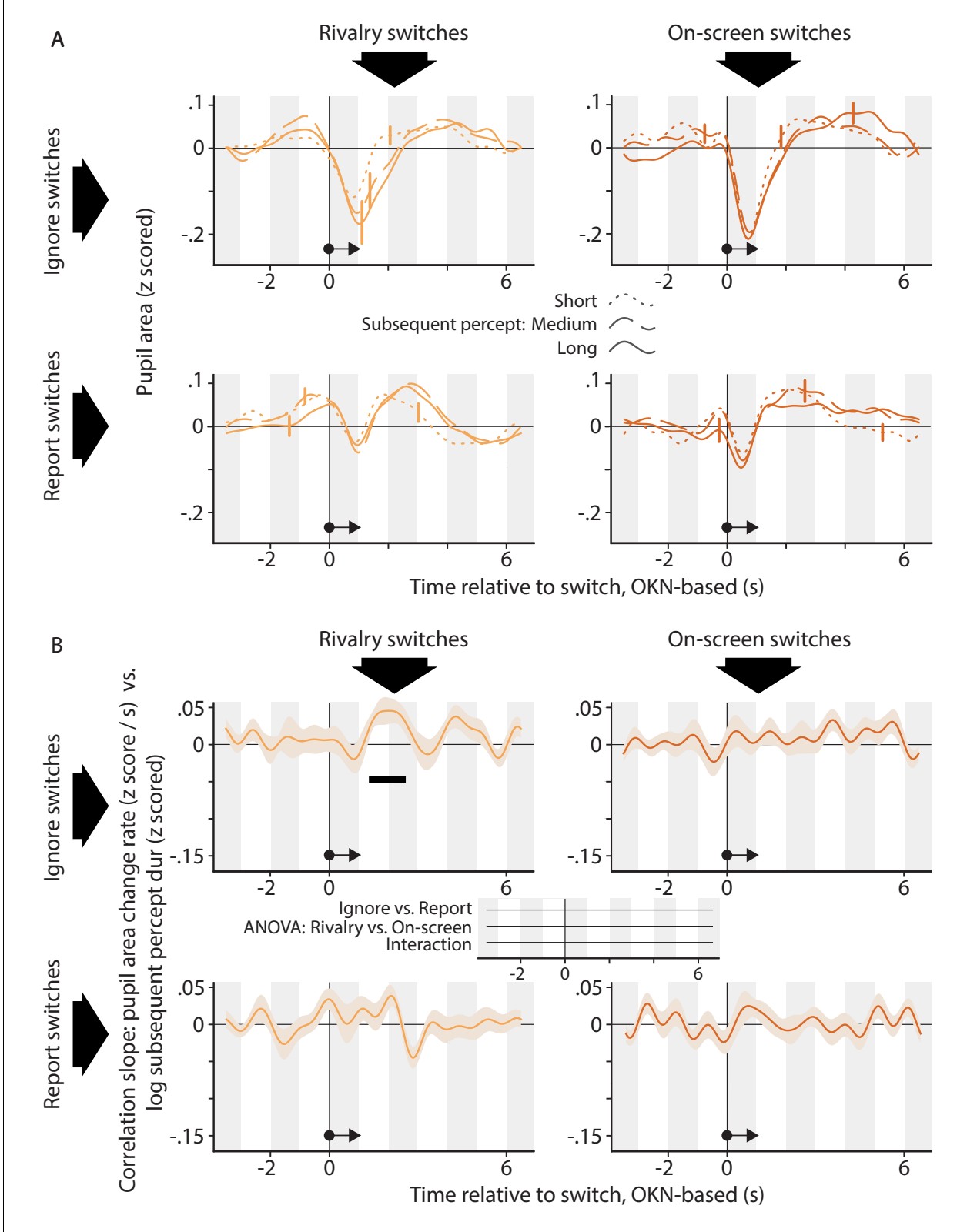

**Figure 6.** The relation between the pupil response associated with a perceptual switch and the duration of the perceptual dominance duration that follows that switch. The organization of this figure is identical to that of *Figure 5*.

that either pupil response component is associated with subsequent dominance duration, is not compelling in our data. A control analysis (*Appendix 1—figure 7*) indicates that the observed relation with preceding dominance duration (*Figure 5*) does not reflect an artefact arising from incomplete separation of overlapping pupil responses to temporally adjacent perceptual switches. On the other hand, that same analysis suggests that what evidence we do observe for a relation with subsequent dominance duration (*Figure 6*) might reflect such an artefact.

## Discussion

We used a combination of both no-report conditions and report conditions, involving both binocular rivalry and on-screen animations, to identify and decompose the switch-related pupil response into two separable components, each associated with different aspects of perceptual switches. The first component is a rapid pupil constriction and re-dilation time-locked to the perceptual switch, irrespective of whether this switch occurs during binocular rivalry or during an on-screen animation. The second is a rapid pupil dilation and re-constriction time-locked to the manual response to the perceptual switch, again irrespective of the nature of the switch. We found a robust dependence of the pupil response on the amount of time elapsed since the perceptual switch that immediately preceded the present one, such that switch-related pupil constrictions are larger following longer dominance periods, both during rivalry and during on-screen replay. We also found some tentative evidence that the pupil response around two seconds following the perceptual switch is related to the duration of the immediately following perceptual dominance period, but this evidence does not warrant strong conclusions.

### Decomposing the pupil response to perceptual switches

As mentioned above, the most robust finding in existing work on this topic has been of a transient pupil dilation (*Einhäuser et al., 2008*; *Hupé et al., 2009*; *Kloosterman et al., 2015b*). Consistent with this, when switch-related pupil responses are discussed in a broader context this is invariably in relation to dilation-linked noradrenergic modulation of cortical function (*Sara and Bouret, 2012*; *Laeng et al., 2012*; *Nassar et al., 2012*). In existing pupillometry studies on multistable perception, the switches were always task-relevant, so we tentatively identify the transient dilation in existing work with the task-related dilation in our present study. Based on results from two such previous studies, we surmise that this dilation is, in part, related to the motor act of reporting itself, but is also related more generally to the task-relevance of the switch events. Both those previous studies included a condition where overt switch reports were omitted, yet where observers did need to attend to *Hupé et al., 2009* or covertly count (*Kloosterman et al., 2015b*) the switches. In both cases, the resulting switch-related dilations were substantially smaller than with overt report, yet not abolished, consistent with some role of task relevance per se. In agreement with authors of previous studies, then, we interpret our task-related dilation response as an example of the pupil dilation that is generally observed in association with elevated cognitive engagement, and that has been linked to increased activity of neuromodulatory centers including the noradrenergic locus coeruleus (*de Gee et al., 2017*; *Laeng and Alnaes, 2019*; *Joshi and Gold, 2020*).

Our *Rivalry Ignore* condition is, to our knowledge, the first to measure pupil responses to task-irrelevant perceptual switches, and to show that these are associated with a pupil constriction, just like switches in our *On-screen Ignore* condition, that is task-irrelevant on-screen switches. We interpret these constrictions in the context of work showing that the pupil may constrict in response to isoluminant changes in visual input, such as changes in color, spatial frequency, or motion content (*Barbur et al., 1992*; *Young et al., 1995*; *Conway et al., 2008*; *Barbur, 2014*). This kind of constriction is similar in size, but opposite in sign, to the engagement-related dilation mentioned above (about 0.1 to 1 mm in diameter; *Slooter and van Norren, 1980*; *Barbur et al., 1992*; *Young et al., 1993*; *Conway et al., 2008*), matching our finding of similar magnitudes for both our positive and our negative rapid response component (but see *Appendix 1—figure 6* for inter-observer differences in the relative magnitudes of the two components). Several lines of evidence suggest a cortical contribution to constrictions in response to isoluminant input transients: these constrictions are virtually abolished by cortical lesions (*Barbur et al., 1992*; *Heywood et al., 1998*), and they are modulated by attention withdrawal and interocular suppression (*Kimura et al., 2014*; *Hu et al., 2019*), as well as by stimulus properties that lack a specific representation outside of cortex (e.g. the

orientation of a viewed face; *Conway et al., 2008*). Indeed, the most developed hypothesis as to what causes such constrictions is that the perturbation of visual cortical responses that results from the visual input change brings about a temporary weakening of tonic inhibition of the midbrain Edinger-Westphal nucleus (*Barbur et al., 1992*; *McDougal and Gamlin, 2008*; *Barbur, 2014*). Since this nucleus drives the iris sphincter muscle, reducing its inhibition would lead to a smaller pupil. It is not certain which pathway would be involved in such putative modulation of tonic inhibition of the Edinger-Westphal nucleus. Among several routes to the Edinger-Westphal nucleus ones involving the pretectal area have been argued not to be critical in this context, based on the observation that the pupil response to isoluminant stimulus changes (but not the pupil light reflex) is preserved in patients with damage to that area (*Wilhelm et al., 2002*). One remaining possibility is that tonic sympathetic inhibition involving the locus coeruleus is interrupted in response to visual cortical events (*Wilhelm et al., 2002*). This option would be consistent with evidence that the locus coeruleus does inhibit the Edinger-Westphal nucleus (*Wilhelm et al., 2002*; *McDougal and Gamlin, 2008*; *Mathôt, 2018*; *Joshi and Gold, 2020*) and that it receives input from cortex (*Samuels and Szabadi, 2008*; *Joshi and Gold, 2020*). Another conceivable route would be via the superior colliculus, which also has the required connectivity with both the Edinger-Westphal nucleus and cortex (*Wang and Munoz, 2015*; *Joshi and Gold, 2020*). One argument against the option involving the locus coeruleus is that the dominant cortical input to that structure comes from frontal cortex as well as the cingulate cortex (*Aston-Jones and Cohen, 2005*; *Sara and Bouret, 2012*; *Joshi and Gold, 2020*), whereas the present switch-related constriction seems to be associated with visual events rather than task execution and, as such, with the back of the cortex rather than the front. It should be mentioned here, however, that some inputs to the locus coeruleus coming from parietal and temporal cortex have also been reported (*Samuels and Szabadi, 2008*; *Joshi and Gold, 2020*). Irrespective of the pathways involved, we propose that the perturbation of visual cortical activity that accompanies rivalry switches (*Logothetis and Schall, 1989*; *Leopold and Logothetis, 1996*; *Tong et al., 1998*; *Polonsky et al., 2000*) leads to a weakening of inhibition of the Edinger-Westphal nucleus, just like visual cortical response perturbations due to isoluminant stimulus changes do. This would explain the switch-related pupil constrictions we report in our rivalry conditions, and it would mean that the switch-related constrictions identified here constitute a new non-invasive index on the visual cortical concomitants of switches in multistable perception. We note that, if it is true that these constrictions reflect a temporary interruption of tonic sympathetic input involving the locus coeruleus, then this renders these constrictions a type of mirror image of the task-related dilations we also observe, as those are thought to reflect a transient elevation of this input.

Although no existing study has shown switch-related pupil constrictions in isolation, previously-reported pupil response shapes do suggest a constriction as one constituent part. As mentioned above, several authors have reported task-relevant switches during perceptual multistability to be linked to a biphasic pupil response composed of an initial dip followed by a peak (*Einhäuser et al., 2008*; *Naber et al., 2011*; *de Hollander et al., 2018*). This temporal order is consistent with the interpretation that those authors have measured a combination of both our rapid constriction (which occurs first, time-locked to the switch) and our rapid dilation (which occurs later, time-locked to the report). Interestingly, reports of such a biphasic response pattern have not been restricted to studies of binocular rivalry (*Naber et al., 2011*; *de Hollander et al., 2018*) but extend to work on other forms of perceptual multistability (*Einhäuser et al., 2008*), suggesting that the constriction arises more generally when perception changes. On the other hand, the most prominent aspect of most published pupil response shapes is the dilation component, suggesting that the constriction component elicited by our particular stimulus is larger than usual.

## Correlations with perceptual dynamics

Existing work has resulted in a somewhat mixed picture of the way in which the durations of flanking perceptual dominance durations are reflected in the switch-related pupil response. *Einhäuser et al., 2008*, in two experiments that used multistable perception paradigms other than binocular rivalry, reported that a more pronounced pupil dilation, centered around 500 ms before the perceptual switch report in one condition, and around 500 ms after in the other, predicted a longer subsequent dominance duration, but *Hupé et al., 2008* questioned that result. *de Hollander et al., 2018* showed results broadly consistent with *Einhäuser et al., 2008*, reporting that a larger pupil during binocular rivalry predicted a longer subsequent dominance duration. However, this correlation was

observed in a slightly different time window again: centered about a second before the report. The most common interpretation of correlations with subsequent dominance duration has been that a switch-related increase of cortical noradrenaline, tied to pupil dilation, would stabilize the newly established perceptual interpretation, thereby delaying the next perceptual switch (*Einhäuser et al., 2008*; *Sara and Bouret, 2012*; see *Kloosterman et al., 2015a* for a conceptually related interpretation of different data). *de Hollander et al., 2018* also observed a correlation with preceding dominance duration: a smaller pupil shortly after the report was linked to a longer preceding dominance period. Those authors interpreted that latter result in terms of the hazard rate of the perceptual switch occurring, rather than in terms of the time interval between switches as such. Specifically, given the roughly gamma-shaped distribution of perceptual dominance durations during perceptual multistability (*Levelt, 1968*; *Borsellino et al., 1972*), the instantaneous probability of a perceptual switch monotonically increases as a function of time since the previous switch. Accordingly, *de Hollander et al., 2018* argued that the relation with preceding dominance duration reflected the degree of surprise associated with the current perceptual switch: low surprise (following long dominance periods) was linked with a smaller pupil as compared to high surprise (following short dominance periods). This interpretation is consistent with a more general body of work on surprise-linked pupil dilations mediated by noradrenaline (e.g. *Preuschoff et al., 2011*), and also with the results of a pupillometry study that specifically manipulated the hazard rate of on-screen switches during replayed perceptual multistability (*Kloosterman et al., 2015b*).

How do our findings on relations with flanking dominance durations compare to the existing literature? Qualitatively speaking there are similarities with the existing work: similar to *de Hollander et al., 2018* we found a smaller pupil (in our case: a stronger pupil constriction) shortly after switches that terminated longer dominance periods (*Figure 5*) and that would, therefore, be less surprising as formalized by the hazard rate. And consistent with both *Einhäuser et al., 2008* and *de Hollander et al., 2018*, we found some anecdotal evidence that a larger pupil predicts a longer subsequent dominance period (*Figure 6*). But there are also aspects of our data that conflict with existing findings and, especially, interpretations. Our data indicate that the main influence of preceding dominance duration is on the switch-related pupil constriction that occurs irrespective of task relevance. This casts some doubt on interpretations centered on surprise and associated noradrenaline release, notions typically related to task-relevant events and to pupil dilations. A more natural interpretation of this aspect of our results is that switch-related pupil constrictions may be subject to a type of adaptation, either of the underlying cortical response or of a component that is closer to the iris musculature. Although we are not aware of any reports of adaptation affecting similar pupil constrictions in the literature, it would explain why our present pupil constrictions are less pronounced during the time period shortly after a previous constriction, as the neural process that mediates constrictions would be less responsive during that period.

A related remark applies to the modest evidence, in our data, that the pupil response may predict the upcoming perceptual dominance duration. To the extent that this effect is real (but see discussion surrounding *Figure 6* and *Appendix 1—figure 7*), it occurs in our *Rivalry Ignore* condition, in which switches were task-irrelevant and in which we did not observe any pupil dilation. This is not consistent with the idea that the magnitude of, specifically, *dilation* predicts upcoming percept duration, nor with the prevailing interpretation in terms of noradrenaline release.

Only in the *Ignore* conditions did we observe significant correlations with preceding percept duration. The data patterns observed in the *Report* conditions, where switch-related constrictions and task-related dilations overlap, did qualitatively match those of the *Ignore* conditions, but showed no significant correlations. One contributing factor here can be that the *Ignore* conditions had more statistical power: the correlated occurrence of both switches (linked to constriction) and key press reports (linked to dilation) in the *Report* conditions means that a larger amount of data is required to obtain a reliable estimate of either individual response component, as compared to the *Ignore* conditions in which the switches occur in isolation. An alternative explanation, that both switch-related constriction and response-related dilation depend on preceding percept duration yet in directions that work against each other in the combined response, is not supported by a control analysis (*Appendix 1—figure 8*). If the explanation is, indeed, to be found in statistical power, then that further underscores the value of our no-report approach to studying the pupillometric correlates of switches in multistable perception.

On the balance, our present data further strengthen the notion (*Kloosterman et al., 2015b*; *de Hollander et al., 2018*) that the perceptual dynamics that precede a perceptual switch have an impact on its pupillary signature, and add to it the finding that this impact is primarily on the switch-related pupil constriction, rather than on the task-related dilation. Related, our findings suggest that explanations in terms of surprise and associated noradrenaline release can, at best, account for part of this impact, and we propose adaptation of the constriction mechanism as a possible additional explanation. With regard to subsequent percept duration, our findings are inconclusive regarding the idea (*Einhäuser et al., 2008*; *de Hollander et al., 2018*) that those can be predicted from the pupil response, but they form no natural fit with the notion that this would have do with switch-related noradrenaline release.

## Pre-switch dilations in the rivalry conditions?

We have centered our interpretations on two response components observed in the wake of perceptual switches: the perception-related constriction and the report-related dilation. But our plots (*Figures 2* and *4*) also appear to show a tendency for the pupil to dilate shortly before the switch in the *Rivalry* conditions, and not in the *On-screen* conditions. We did not center our interpretations on this tendency because its significance level does not quite reach our chosen alpha of 0.01 in our analyses. Still, the obtained significance levels are sufficient to consider whether this pupil signal may be real: $p=0.04$ for the positive rate around 1 s before the switch in *Figure 2B*, *Rivalry Report*; $p=0.013$ for the positive rate between about 2 s and 1 s before the switch in *Figure 4B*, top plot, although in both cases the *Rivalry* versus *On-screen* comparison, (from the ANOVA in *Figure 2B* and from the paired t-test in *Figure 4B*) reaches only $p=0.17$ for the time window in question. Such a pre-switch dilation, if real, would be consistent with measurements in other experimental domains that suggest elevated cognitive engagement around the time of perceptual switches during perceptual multi-stability, even as compared to on-screen replay conditions (e.g. frontoparietal fMRI BOLD signals, *Lumer et al., 1998*; various EEG markers, *Kornmeier and Bach, 2012*). Discussion as to the role of the underlying brain signals in supporting the observer's perception and behavior is ongoing (*Brascamp et al., 2018b*), and the fact that our data seem to show a stronger pre-stimulus dilation in the *Rivalry Report* condition than in the *Rivalry Ignore* condition (*Figure 2*) could be seen as evidence for a role in translating the perceptual experience into a behavioral response.

## Limitations of the OKN-based method of identifying switches

*Figure 1C–E* show good correspondence between switches identified using the OKN-based algorithm and both manual switch reports and on-screen switch events. Nonetheless, our algorithm appears to overestimate the number of switches for slow switchers, and to underestimate it for fast switches (*Figure 1C*). One possibility is that the OKN-based algorithm specifically tends to miss switches that are closely spaced in time, but a control analysis argues against that possibility (*Appendix 1—figure 4*). Although this leaves us without a hypothesis as to the specific nature of the switches that the algorithm misses, we do propose the following general interpretation of the observed data pattern. Our OKN-based algorithm may have both a non-zero, and fixed, false alarm rate (i.e. a certain number of spurious switch events is marked per unit time), and also a hit rate that is lower than 100% (i.e. a certain proportion of actual switches is not marked). The impact of the former factor, which leads to an overestimation of the switch rate, may be relatively constant across observers, whereas the impact of the latter factor, which leads to an underestimation, is bound to be larger for observers who experience more switches. This combination of factors, therefore, may explain the observation that a net overestimation of the switch rate for slow switchers gives way to a net underestimation for fast switchers (*Figure 1C*). While these considerations mean that the OKN-based algorithm is not perfect (and that it may be improved by incorporating some recently proposed analysis choices; *Aleshin et al., 2019*), this does not importantly affect our conclusions as long as there is a close association between switches as marked by the various methods, which there is (*Figure 1D–E*). In other words, for our purposes the critical requirement is that the OKN-based algorithm marks time points that are strongly associated with perceptual switches; not that it marks zero spurious time points nor that it marks a time point for every single perceptual switch. This is especially true because our key analyses concern comparisons between conditions while keeping the switch-identification method constant; not comparisons between identification methods.

## Conclusion

The application of pupillometry methods in the context of multistable perception holds promise as an approach to studying perception and its neuromodulatory dependencies, both because the pupil non-invasively informs about transient noradrenergic activity accompanying perceptual switches, and because such activity may be reflected, on a slower timescale, in the spontaneous dynamics of the perceptual cycle. We demonstrate that the inclusion of *Ignore* conditions, in which switches are stripped of their cognitive significance, allows for a more incisive characterization and interpretation of switch-related pupil responses. Taking this approach we provide evidence that this response — hitherto treated as a unitary signal — is composed of two overlapping but separable components, each associated with a different perceptual or cognitive process. While one is a task-related dilation component that is plausibly associated with a transient rise in noradrenergic activity stemming from brainstem arousal systems, the other is a constriction component that has likely contributed to pupil signals reported in the literature, and that may signify a temporary release from inhibition of the Edinger Westphal nucleus. Given that this release of inhibition would be a consequence of an altered visual cortical response to visual input, the constriction component arguably provides a novel and easily accessible index of the visual cortical response change that marks perceptual switches. As such, this work offers insight into the neural processes involved in perceptual switching, as well as providing a new methodological and conceptual reference point for future pupillometry work on this topic to fully deliver on its promise.

## Materials and methods

### Observers

Observers were recruited from the Michigan State University undergraduate and graduate student population (age range 18–30 years). All were naive to the purposes of the investigation. The study protocol was approved by the Michigan State University institutional review board, and observers received financial compensation for their participation. During their first visit to the lab observers received informed consent and were familiarized with the stimulus during a colloquial interaction. On that occasion, the experimenter verified that the observer experienced perceptual alternations and that the eye tracker got a stable read of the observer's pupils. Based on these criteria, 26 observers were enrolled in the experiment proper. After initial data analysis two observers were excluded from further analysis because they reported an excessive amount of perceptual mixtures (*Appendix 1—figure 9*), which we deemed undesirable given our interest in switches between exclusive percepts. This left 24 observers whose data are reported in the main text.

### Stimulus and task

The stimulus consisted of dots (radius 0.17 dva, density 2.7 dots/dva$^2$) randomly placed within a round aperture (radius 3.9 dva) and moving either leftward or rightward at 4.1 dva/s on a gray background (34.5 cd/m$^2$). Half of the dots of a given color were lighter than the background (62.8 cd/m$^2$) and half were darker (19.0 cd/m$^2$). One of the colors, cyan, was created by setting the screens' blue and green channels to the same luminance and turning off the red channel. The other color, magenta, was created by setting the red and blue channels to the same luminance while turning off the green channel.

The stimulus was surrounded by a fusion aid that consisted of a coarse random pixel array (pixel side 0.72 dva) with an equal number of dark (69.1 cd/m$^2$) and light pixels (129 cd/m$^2$), overlaid by a small black frame (side 15.5 dva; 2.9 cd/m$^2$) and a larger white frame (side 18.6 dva; 336 cd/m$^2$). The pixel array itself filled a square area (side 23.2 dva) except for a circular area (radius 7.7 dva) at its center. Observers viewed the stimuli on two separate computer monitors (one for each eye) via a mirror stereoscope designed to be compatible with video-based eye trackers (*Qian and Brascamp, 2017*; *Aleshin et al., 2019*).

Each observer completed two blocks for each condition, so eight blocks in total. The blocks were spread out across multiple visits to the lab, typically between two and four. The observer's eyes were tracked binocularly at 1000 Hz using an Eyelink 1000 Plus video-based eye tracker (SR Research, Ottawa, Canada). During each block the observer first performed a procedure in which he or she visually aligned two frames shown in alternation, each on a different monitor. The

corresponding screen coordinates were stored to present the two eyes' stimuli at corresponding visual locations during the experiment. After an eye tracker calibration the observer then completed 12 trials of 60 s each, all for the same condition. Dot color and dot direction were yoked. On half of the rivalry trials, randomly assigned, dots of a given direction and color were shown to one eye; on the remaining trials they were shown to the other eye. A trial's initial dot positions were randomly determined at the start of the trial, independently for each trial. Between trials the observer was allowed to pause as needed, and he or she performed a drift correction procedure before starting the next trial. If the tracker did not get a stable reading during this procedure, or if gaze direction had drifted more than six dva since calibrating, a new calibration procedure was completed before starting the next trial. The tolerance of this drift correction procedure was very large (six dva) for an experiment aimed at measuring absolute gaze direction, but this approach proved to be efficient in this case, where pupil size and gaze displacement were important but absolute gaze direction was not.

During the *Report* conditions observers used three keyboard keys to indicate each trial's initial percept as well as any moments at which perception changed. Two of the buttons corresponded to exclusive leftward or rightward motion, respectively, and the third button corresponded to mixture percepts. During *Ignore* conditions observers pressed one keyboard button each time they identified a so-called 'dot size probe'. These were occasions where all dots, across both eyes, simultaneously shrank over the course of 250 ms and then immediately grew back to their original size during another 250 ms. At the start of each block, this size change was set to 20% (i.e. a shrinkage down to 80% of the normal size), but it was altered during the experiment using a staircase procedure: for each missed probe the size change was multiplied by 1.1, and for each correct detection it was divided by 1.1. Across observers the average staircase convergence point was 12.1%. The interval between consecutive probes was drawn randomly from a uniform distribution between 3 s and 8 s.

During the *On-screen* conditions, the same display was shown on both eyes' screens, with either only cyan dots going in one direction, magenta dots going in the other direction, or a mixture of the two. During these mixture periods the circular stimulus aperture was split midway into a top and a bottom half, and the two halves each showed dots of a different color, and going in a different direction. This is not a realistic rendering of perceptual mixtures during rivalry, which do not typically involve a clean split between the two eyes' dominance regions. In previous work we have attempted more realistic on-screen mimics (*Knapen et al., 2011*; *Brascamp et al., 2018a*) but we are not aware of ones that convincingly simulate rivalry's perceptual experience, and we see no reason why such a more realistic mimic might have importantly altered our present results.

During each block of the *On-screen* conditions, we replayed perception as reported during the observer's most recent block of the *Rivalry Report* condition, that is using the percept timing reported there while assuming a fixed reaction time of 500 ms. (The positions of individual dots, however, were not copied from the earlier *Rivalry Report* trials but, instead, determined randomly at the start of every trial.) This approach meant that a *Rivalry Report* block had to precede any *On-screen* block for a given observer. To still minimize any role of time or experience, for each observer the first four blocks included exactly one block of each condition, in random order while heeding the constraint specified above, and the last four blocks again included all four conditions but in reverse order.

## Sample size

Sample size was not based on an explicit power analysis. The amount of data per observer-condition was adjusted upward toward its final value on the basis of pilot experiments that showed a lack of robust switch-related pupil responses within observers at smaller values yet stable pupil responses for many observers at the final value. For our number of observers (24 whose data were included), we chose a value that was above the high end of the observer numbers reported across relevant published studies, given that we wished to replicate and extend upon the pupil responses reported in those studies (six observers per experiment in *Einhäuser et al., 2008*; 10 and 14 observers per experiment in *Hupé et al., 2009*; 22 and 19 observers per experiment in *Kloosterman et al., 2015b*; nine observers in *de Hollander et al., 2018*). The number of perceptual switches underlying our switch-related pupil curves averaged about 430 per condition per observer (about 10,000 switches per condition in total across observers).

## Data analysis

### Percept dynamics inferred from key presses

In extracting percept sequences from key presses, we ignored all key presses that repeated the previous one. For the purposes of on-screen replay all transition periods were registered at their manually reported duration, including so-called 'return transitions' in which perception changed from one exclusive percept to a mixture and then back to the same exclusive percept again (*Mueller and Blake, 1989*). When it comes to OKN-defined switches, on the other hand, because of the difficulty in accurately delineating perceptual mixture periods based on eye movements (although see *Aleshin et al., 2019*; *Qian and Brascamp, 2019* for progress in that direction) those were considered instantaneous and, by definition, between two different percepts (see next section). Therefore, in those instances where either key-defined switch timing or on-screen switch timing was compared to OKN-defined switch timing (either directly in *Figure 1C–E*, or indirectly in *Appendix 1—figure 5*) we placed an instantaneous switch moment midway each key-defined or on-screen mixture period that separated two different percepts, and we ignored return transitions.

### Percept dynamics inferred from eye movements

We inferred percept dynamics from eye movements using an approach similar to authors of previous studies (*Naber et al., 2011*; *Frässle et al., 2014*; *Aleshin et al., 2019*). The first analysis steps were aimed at obtaining a clean gaze position signal. We first split the gaze samples into contiguous sequences of 5 samples or more, thus removing all periods without signal as well as sample sequences that are too short for applying our subsequent saccade detection algorithm. This algorithm, from *Engbert and Mergenthaler, 2006*, was applied to each contiguous sample sequence to identify saccades on the basis of gaze displacement velocity. Here, minimum saccade duration was set to 6 ms (six samples), and the velocity threshold was set to six standard deviations (using median-based standard deviation as described in the original study). Saccades were initially identified independently for each eye, after which temporally overlapping saccades from the two eyes were marked as binocular, and were assigned whichever of the two eyes' saccade start times was earlier, and whichever end time was later. To identify eye blinks we relied on the Eyelink preprocessing software, which marks all periods of signal loss as blinks, separately for each eye. Blink events were then combined across eyes in the same way as just described for saccades. After saccades and blinks had been identified in this fashion we averaged gaze position across the two eyes and we replaced all samples that were closer than 20 ms to a saccade or closer than 50 ms to a blink. In particular, gaze positions for sample sequences that were separated by such samples were collated such that any gaze displacement during those samples was set to zero. These processing steps led to data like those depicted in the center plot of *Figure 1B*.

After the gaze position signal had been cleaned in this fashion, the next processing steps were aimed at identifying perceptual switch moments. For this purpose, we slid a window of 750 ms width over the cleaned gaze position signal in steps of 38 ms. At each step, we assigned to the time point at the center of the window a value that quantified the direction of gaze displacement within the window. In particular, we first fit a linear curve to the vertical gaze position vs. time data within the window, and another linear curve to the horizontal gaze position vs. time data. The arctangent of the two slopes quantified the gaze displacement angle on the screen within that time window. Because we were interested in eye movements in response to horizontally moving stimuli we then computed the cosine of this angle, which ranges from −1 for due left gaze displacement to +one for due right gaze displacement. During periods without a signal (due to actual eye closure or due to eye lock interruptions of a different nature) gaze displacement had been artificially set to zero (see above). Because those periods were sometimes of non-negligible duration our estimates of gaze displacement direction were sometimes unreliable near periods marked as blinks. For this reason, all time points that fell in a time window between 250 ms before the start of a blink and 400 ms after the end of that blink were assigned the average gaze displacement value computed across the 100 ms immediately before, and the 100 ms immediately after that time window. These processing steps together led to data like those depicted in the bottom plot of *Figure 1B*. On the basis of these data, all time points where the cosine of the gaze displacement angle was larger than 0.85 were assigned to one percept, and all time points where that value was smaller than −0.85 were assigned to the other. As the final analysis step perceptual switch moments were marked as all moments that

lay midway two adjacent time periods that had been assigned to opposite percepts, with the exception that switches were not marked if they produced a perceptual dominance episode that was briefer than 500 ms.

## Pupil preprocessing

The pupil area signal was first averaged across eyes. For each blink (identified by the Eyelink software and then combined across eyes as specified above), we then replaced pupil size during the interval from 50 ms before the blink to 85 ms after the blink with values that linearly interpolated between the average pupil sizes during the 50 ms periods that preceded and followed that interval. We observed a tendency for pupil size to slowly drift, usually downward, over the course of each 60 s trial; a tendency reported previously (*Knapen et al., 2016*), although the opposite has been observed as well (*Binda et al., 2017*). We therefore followed authors of previous work (*Van Slooten et al., 2017*) by fitting and then subtracting an exponential curve to each trial's pupil size data (after interpolating blinks). Here we constrained the fitted time constant to values slower than 10 s to ensure that this step captured slow drift rather than transient pupil changes early in the trial. Each trial's residual was then low-pass filtered using a third order Butterworth filter with a cut-off frequency of 6 Hz, and z-scored. For all samples that fell between trials the pupil size was set to 0. The data were then downsampled to 10 Hz and concatenated across conditions.

## General linear models

We used a general linear model (GLM) approach to evaluate the temporal relation between switch moments as identified using different methods (*Figure 1D–E*) and to evaluate pupil responses associated with specific events.

*Figure 1D and E* provide estimates of the temporal relation between switch events as identified by distinct methods. From the raw data, it is not always clear which event from one method (e.g. based on key presses) corresponds to which event from a comparison method (e.g. based on OKN), for instance because one method might miss an event or mark a spurious event where the other method does not, or because switch events may sometimes follow each other too closely to confidently match up event pairs across methods. For this reason, we did not attempt to explicitly identify pairs of corresponding switch events between different methods to then compute the time delay between the two. Instead, we took a deconvolution approach, which does not require one to explicitly identify such correspondence. For each observer, we took the list of switch moments as marked using the method specified in a plot's y-axis label, and converted it to a time-varying signal sampled at 10 Hz by entering a one at every time step that contained a switch and a 0 elsewhere. We then ran a GLM deconvolution analysis that combined that time-varying signal with the switch event times as marked by the method specified in the corresponding x-axis label. We did this for each switch direction separately (percept A to B and vice versa) and averaged the two resulting deconvolution curves for each observer. The analysis was run using the FIRDeconvolution package (*Knapen et al., 2016*; *Knapen and de Gee, 2016*) using a deconvolution time step of 100 ms, and concatenating the data across all trials (and both blocks) of a given condition. For each of the curves in *Figure 1D–E*, we used a deconvolution interval of 3 s long, but its position was different for each curve (*Figure 1D* OKN-based: −1 s to 2 s; *Figure 1D* reported: −1.5 s to 1.5 s; *Figure 1E*: −2 to 1 s). To compute the estimates of hit rate (HR) shown in the bar charts of *Figure 1D–E*, we took the area under each curve within a 1.5 s window that, based on visual inspection, contained the bulk of the probability mass (*Figure 1D* OKN-based: −0.5 s to 1 s; *Figure 1D* reported: 0 s to 1.5 s; *Figure 1E*: −1.5 s to 0 s). To compute the estimates of false alarm rate (FA) in the same plots, we took the area under each curve during the remainder of the 3 s deconvolution interval.

For the pupil analyses, we used a different type of GLM approach that aims to reconstruct the event response using a set of basis functions (*Friston et al., 1998*). This approach strikes a balance between deconvolution analyses and GLMs that are based on a standard response function (a pupil response function in this case; *Hoeks and Levelt, 1993*; *de Gee et al., 2014*; *Denison et al., 2020*). The former have a high degree of flexibility in terms of the response shapes they can reconstruct, at the expense of many degrees of freedom (as many as there are time points in the reconstructed response). The latter have few degrees of freedom (e.g. only a scaling parameter) at the expense of flexibility. The present approach based on basis functions is intermediate: the pupil response here is

modeled as a weighted sum of functions from a series (e.g. a Fourier series or a Taylor series), and the number of functions included determines the flexibility and degrees of freedom. In our case we used the ResponseFitter class from the nideconv package (*de Hollander et al., 2018*; *de Hollander and Knapen, 2018*) to fit the first terms of a Fourier series to the pupil signal. We fitted both an offset and the number of sines and cosines needed to capture fluctuations at a frequency of 1 Hz and slower, which meant 21 terms for most regressors, for which the fitted time window ran from 3.5 s before to 6.5 s after the event. For the blink regressors (fitted between −0.5 and 7.5 s) and the saccade regressors (fitted between −0.5 and 4.5 s) it meant 17 and 11 terms, respectively. We independently performed this analysis on the preprocessed pupil time series itself, and on its derivative (*Figures 2* and *4* show the result of both in separate panels).

For the analyses of *Figure 2*, the regressors in our design matrix were based on the following: OKN-based switches (for each condition separately), trial start events, saccades, and blinks. For the *Ignore* conditions, we furthermore included key presses (which were in response to dot size probes) and unreported dot size probes (defined as those probes that were not followed within 2 s by a key press). For the analyses of *Figure 4*, the regressors were constructed using OKN-based switches across only the two *Rivalry* conditions combined, OKN-based switches across only the two *Replay* conditions combined, key presses across all conditions, trial start events, saccades and blinks. For all pupil analyses, the regressors (except the trial start regressors) excluded those events that occurred so close to the start or end of a trial that the modeled time window would extend beyond the trial period. For the saccade regressors, we merged pairs of saccades (identified as described above) that were fewer than 100 ms apart because of the impression, based on visual inspection of the gaze traces, that these instances usually concerned single square-wave intrusion events or saccadic pulse events (*Abadi and Gowen, 2004*). For the blink regressors, we excluded events that the Eyelink had marked as blinks but that were shorter than 130 ms or longer than 900 ms, because those were more likely to reflect signal loss for reasons other than blinks (*Kwon et al., 2013*). For *Figures 5* and *6*, we used the same basic design matrices but added covariates to the switch regressors. These covariates were formed by the OKN-based percept durations (log transformed and z-scored) that preceded or followed the switches.

## Statistics

All statistics were performed at the across-observer level. In particular, we computed pupil response curves for each observer individually (using the methods described in the previous sections; see *Appendix 1—figure 6* for such observer-level curves), and then performed statistical tests (t-tests and ANOVAs) on the across-observer distributions of pupil response values. Performing such tests for each individual time point within the analyzed pupil response period would result in a multiple comparison problem. Moreover, correcting for the large number of comparisons would not be straightforward because of the non-independence of adjacent time points within a given response curve (i.e. the statistical tests at different time points would not be independent). We therefore followed an approach that has been proposed for this type of situation, and that centers on cluster-level significance rather than per-timepoint significance (*Bullmore et al., 1999*; *Maris and Oostenveld, 2007*). Conceptually this means the following. When using per-timepoint statistics a *p*-value would be associated with a specific time point, and would represent the probability, under the null hypothesis, of observing an across-observer distribution of response values that are as extreme as the observed set of values at that time point. In the case of cluster-level statistics, on the other hand, a p-value is associated with a cluster of time points that is contiguous across time, and it represents the probability, under the null hypothesis, of observing a cluster, contiguous across time, of extreme per-timepoint sets of values that is as large as the observed cluster. More specifically, we computed cluster-level Monte Carlo p-values using Bullmore's cluster mass test (*Bullmore et al., 1999*; *Maris and Oostenveld, 2007*). To this end, we first performed conventional tests (t-tests or repeated measures ANOVAs, in different cases) for each time point separately, and formed clusters out of groups of adjacent time points that all had p<0.05 (two-tailed) and the same sign of effect. For each cluster, we then computed the 'cluster mass', that is the sum of all time points' test statistics (t values or F values, depending on the test). We then performed 1000 iterations of a permutation procedure to establish the probabilities of cluster mass values at least as extreme as the ones observed. For the repeated measures ANOVAs and paired t-tests each iteration involved randomly

assigning the observed data to conditions for each observer independently *Maris and Oostenveld, 2007*; for one-sample t-tests, each iteration involved randomly inverting or not inverting the sign of the observed data for each observer independently (*Nichols and Holmes, 2002*; their example 3). On each iteration, we computed cluster mass values based on the randomized data by applying the procedure described above and stored the most extreme of those values, thus forming a permutation distribution of 1000 values. Each cluster identified in the actual, non-randomized, data was then assigned a Monte Carlo p-value equal to the proportion of the permutation distribution that was more extreme than the cluster's observed mass. All clusters with a Monte Carlo p-value smaller than 0.01 were considered significant.

## Acknowledgements

The authors thank Matthew Zadel for collecting data for the control experiment of *Appendix 1—figure 3C*.

## Additional information

### Funding
No external funding was received for this work.

### Author contributions
Jan W Brascamp, Conceptualization, Resources, Data curation, Software, Formal analysis, Supervision, Investigation, Visualization, Methodology, Writing - original draft, Project administration, Writing - review and editing; Gilles de Hollander, Methodology, Writing - review and editing; Michael D Wertheimer, Ashley N DePew, Investigation; Tomas Knapen, Conceptualization, Methodology, Writing - review and editing

### Author ORCIDs
Jan W Brascamp (iD) https://orcid.org/0000-0001-7955-5479
Gilles de Hollander (iD) https://orcid.org/0000-0003-1988-5091
Ashley N DePew (iD) http://orcid.org/0000-0002-8398-7319
Tomas Knapen (iD) http://orcid.org/0000-0001-5863-8689

### Ethics
Human subjects: Informed consent, and consent to publish, was obtained, and all research was approved by Michigan State University IRB, and executed in accordance with the Michigan State University IRB guidelines. The MSU IRB protocol number associated with this work is IRB# 17-996.

### Decision letter and Author response
Decision letter https://doi.org/10.7554/eLife.66161.sa1
Author response https://doi.org/10.7554/eLife.66161.sa2

## Additional files

### Supplementary files
• Transparent reporting form

### Data availability
The raw data associated with this study are available from https://datadryad.org/ (https://doi.org/10.5061/dryad.41ns1rncp) Analysis code associated with this study is available from GitHub (https://github.com/janbrascamp/Pupils_during_binocular_rivalry (copy archived at https://archive.software-heritage.org/swh:1:rev:b5412dcf53f26a0aa96c30fb3f80511a356f7d14)).

The following dataset was generated:

| Author(s) | Year | Dataset title | Dataset URL | Database and Identifier |
|-----------|------|---------------|-------------|------------------------|
| Brascamp JW | 2021 | Separable pupillary signatures of perception and action during perceptual multistability | https://doi.org/10.5061/dryad.41ns1rncp | Dryad Digital Repository, 10.5061/dryad.41ns1rncp |

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

# Appendix 1

## Artefacts from the dot size probe manipulation

In the *Ignore* conditions, observers were asked to do an alternative task to which perceptual switches were irrelevant. This task consisted of reporting transient, small, changes in the size of the stimulus dots (see main text for details). Because the stimulus dots did not have the same luminance as the background, these transient size changes entailed a change in overall light levels and it is possible that this impacted pupil size. It is unlikely that these putative pupil light responses play an important role in our assessment of switch-related pupil responses because the dot size probes occurred at randomly determined moments and therefore were not temporally related to the switch events (this is especially true in the *On-screen Ignore* condition where the switch events also occurred at computer-generated moments; in the *Rivalry Ignore* condition, it is conceivable that dot size probes causally influenced the occurrence of perceptual switches). Nevertheless, we performed control analyses to evaluate the potential impact of the dot size changes on our findings.

First, we examined whether unreported dot size changes resulted in any detectable pupil size change, because the unreported ones allow us to isolate the effect of the luminance changes themselves. *Appendix 1—figure 1A* shows the pupil response to those events (top panel: pupil area; bottom panel: pupil area change rate), as estimated using a general linear model applied to the concatenated data across all conditions, that also includes all other important event types (perceptual switches within each condition, reported dot size changes, trial start events, blinks and saccades). The rate of pupil size change is not significant at any point within the evaluated time window, suggesting that any luminance change associated with these events is small enough to have negligible effect on pupil size. It must be added, however, that dot size probes that the observers do report, result in a substantial pupil dilation (see *Appendix 1—figure 7*), as would be expected from that type of task relevant event. Moreover, the magnitude of the dot size probes was controlled using a staircase procedure (see main text), and it is plausible that the non-reported ones involved particularly small luminance changes. To more conclusively assess the role of these dot size events in our main findings, we repeated the analysis of switch-related pupil responses in our *Ignore* conditions, now only using switch events that were at least 1.5 s removed from the nearest dot size event (the remaining switch events were included as a nuisance regressor). The results of this analysis (*Appendix 1—figure 1B*) show that the estimated switch-related pupil responses have all the relevant properties of the ones estimated in the main text when including all switch events irrespective of dot size changes (main text, *Figure 2*), providing further evidence against the notion that pupil size changes in response to the dot size probes are relevant to our main conclusions.

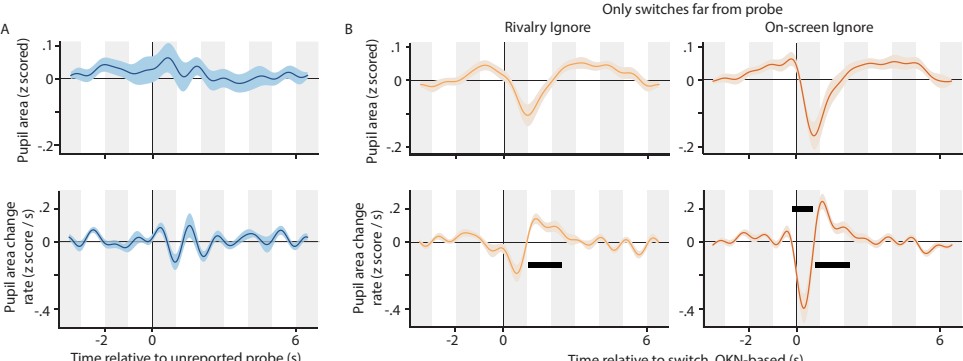

**Appendix 1—figure 1.** The impact of dot size probes on pupil size. (**A**) Pupil area (top) and pupil area change rate (bottom) during the time period surrounding unreported dot size probes. The pupil change rate is not significant at any point in this period. (**B**) The switch related pupil response during the *Ignore* conditions, using only switches that were at least 1.5 s removed from the nearest dot size probe.

## Dependence on the direction of the perceptual switch

Perceptual switches in our design entailed a perceptual transition between two different colors. Although we designed the stimulus for both colors to be matched in luminance, it is possible that the match in (effective) luminance was imperfect. In the *Rivalry* conditions, moreover, perceptual switches entailed a perceptual transition between two different screens and between two different eyes. Again, the (effective) luminance match between the screens, and between the eyes, may have been imperfect. While all our estimates of switch-related pupil responses involved aggregating over all switch 'directions' (i.e. toward either color, screen, and eye), this does not rule out the possibility that some part of those responses is due to a putative imbalance in (effective) luminance. For instance, constrictions in response to a luminance increment are generally faster than dilations in response to the opposite luminance decrement (e.g. *Laeng and Alnaes, 2019*) so the two would not average out. In this control analysis we re-examine the switch-related pupil response during the *Ignore* conditions (which isolate the pupil response that is associated with the perceptual change as opposed to task performance), now separately assessing the response for different directions of the perceptual switch. *Appendix 1—figure 2A* shows the result for both the *Rivalry Ignore* condition (left pair of plots) and the *On-screen Ignore* condition (right pair of plots) when separating by the two colors, each designated by a separate curve.

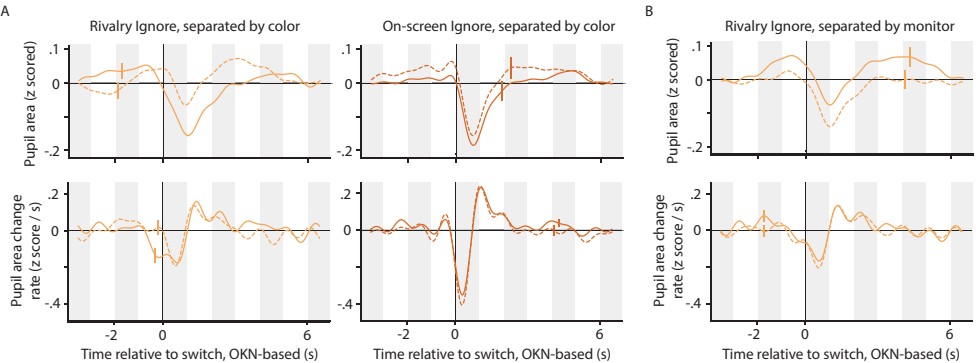

**Appendix 1—figure 2.** Switch-related pupil responses for switches in various 'directions'. (**A**) Switch-related pupil responses in both *Ignore* conditions, separated by the color/motion direction of the (perceived) stimulus following the switch (motion direction was yoked to color in our experiment). Within each plot each line corresponds to one of the two colors. (**B**) Switch-related pupil responses in the *Rivalry Ignore* condition, separated by the eye/monitor associated with the perceived stimulus following the switch.

The pupil responses for both colors are overall comparable to the ones shown in the main text when aggregating across colors (main text *Figure 2*), and at no point is the rate of pupil size change different between the two colors (the numerical rate difference immediately before the switch in the *Rivalry Ignore* condition is associated with a *p*-value of 0.12). *Appendix 1—figure 2B* shows the results of an analogous analysis that separates by screen (or, equivalently, eye) in the *Rivalry Ignore* condition. Again, both curves show a similar data pattern, and there is no statistical difference in rate of change at any time. Taken together, these results indicate that our main results are not impacted in a critical way by differences in pupil response between the opposite switch directions.

## Artefacts from blinks, saccades, and gaze angle

Blinks and saccades are associated with pupillary signals of several kinds. When, as is the case here, one uses a video-based eye tracker there is an apparent change in pupil size when the eyelid (partly) covers the pupil during a blink, as well as when gaze angle changes (due to a saccade or otherwise) and the projected size of the pupil within the camera image is altered as a result (*Gagl et al., 2011*; *Laeng and Alnaes, 2019*). In addition, a physiological pupil response has been observed during the seconds that follow a blink or a saccade (*Hupé et al., 2009*; *Knapen et al., 2016*). Unlike experimental designs that involve discrete and short trials, our design did not allow trials with a blink to be discarded, and our OKN-centered approach obviously led to substantial changes in gaze angle and a

large number of saccades during data collection. We need to consider, therefore, how blinks, saccades, and gaze direction may have influenced our results.

Before going into control analyses and a control experiment that we performed in this context, we will discuss the plausibility of an important influence of ocular events given the particulars of our findings and analysis approach. First, the idea of an important influence of this kind on our basic finding of two switch-related pupil response components—a constriction and a slightly later dilation—is rendered less convincing by the good match with published results. As discussed in the main text, several previous studies have reported that the pupil response to task-relevant switches in multistable perception, although dominated by a dilation, includes an earlier constriction, as well. None of those studies employed a stimulus designed to induce OKN, and one of the studies (*de Hollander et al., 2018*) specifically excluded switches from analysis if they closely followed a blink. As such, an important role for OKN or blinks in this main result is unlikely. Aside from this general consideration, our analysis approach included steps to alleviate the influence of saccades and blinks. Pupil size in the period during and immediately surrounding blinks was discarded and interpolated, addressing the issue of artefactual pupil constriction due to eyelid closure. In addition, blink events and saccade events were both included as regressors in our linear models, so the bulk of the physiological pupil response that follows those events is captured by those regressors and thereby separated from the switch-related responses. Exactly what proportion is captured depends on the suitability of the assumptions that come with a linear model approach (e.g. the assumption of linearity), but we are reassured by the good correspondence between the blink-related and saccade-related responses that we estimate (*Appendix 1—figure 3A*), and ones observed previously (*Hupé et al., 2009*; *Knapen et al., 2016*).

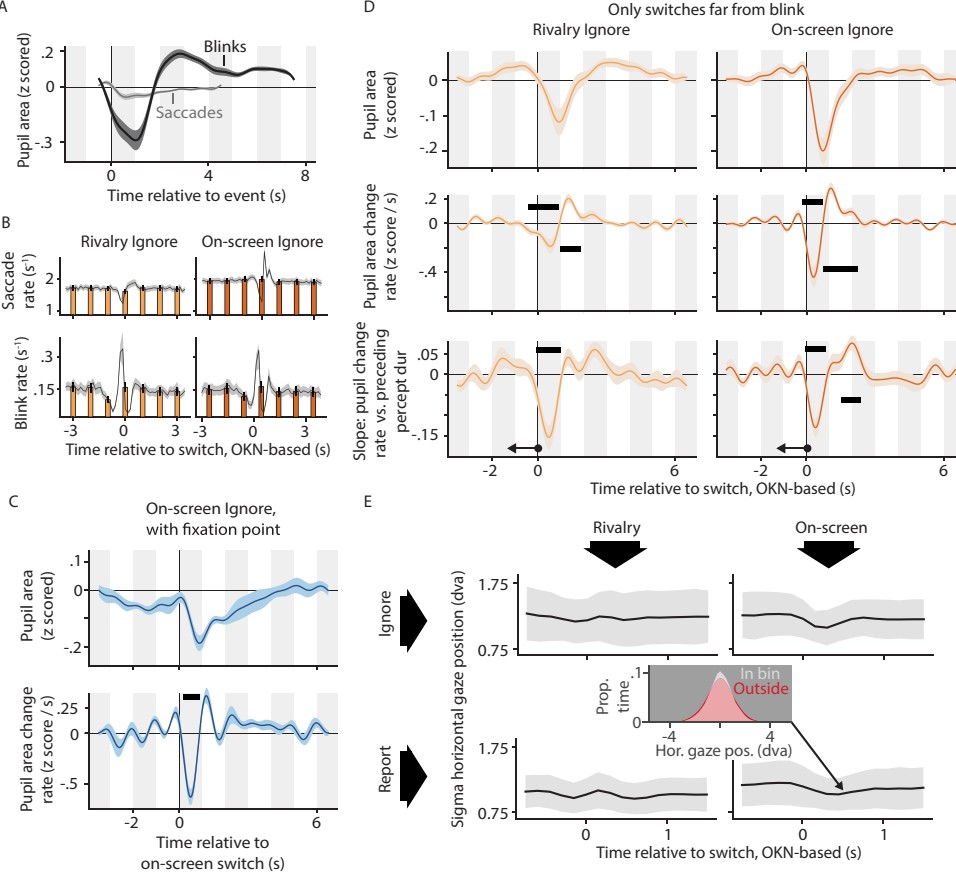

**Appendix 1—figure 3.** The role of blinks and saccades in shaping switch-related pupil responses. (**A**) All linear models underlying the main text figures included designated regressors for blink events and saccade events. Pupil response shapes linked to those events are shown here for one of those

*Appendix 1—figure 3 continued on next page*

*Appendix 1—figure 3 continued*

models (these results are highly similar across the various models). (**B**) Saccade rate (top plots) and blink rate (bottom plots) during the time period around the perceptual switch in two of our conditions, assessed both at a fine time scale (black curves and gray confidence intervals) and at a coarse scale of 1 s per bin (bar charts). (**C**) Pupil response near the moment of on-screen switches in a variant of the *On-screen, ignore* condition in which we removed a circular region (radius: 0.72 dva) from the center of the stimulus and included a binocular fixation point in the middle, instead. (**D**) Re-analysis of the main dataset for two conditions, now focusing exclusively on switches that are at least 1.5 s removed from the closest eye blink. Both the switch-related pupil response (top and center plots) and its dependence on the previous percept duration (bottom plots) are similar to the main result, obtained using switches irrespective of nearby blinks (main text *Figures 2* and *5*). (**E**) The width of the horizontal gaze position distribution as a function of time relative to perceptual switches. For each time point each plot shows the width of the distribution of all on-screen gaze positions recorded within a 200 ms window centered on that time point relative to a perceptual switch. The inset shows, for one such time point, the distribution of gaze positions recorded within this window (in gray) and the distribution of gaze positions recorded outside of the window (in red; pink is the area of overlap). For the analysis of figure E we took the average gaze position across an entire 60 s trial to correspond to the center of the screen. This is because our experiment design was optimized for assessing pupil size and gaze direction change (i.e. OKN); not for assessing absolute gaze direction. In particular, our drift correction procedure at the start of each trial had a rather large tolerance for absolute gaze errors, in order not to interrupt the experiment with additional calibration procedures (see main text Materials and methods, section Stimulus and task). Unless otherwise stated, all conventions in this figure are the same as those introduced in main text *Figures 2*, *4*, *5* and *6*.

We performed several control analyses and a control experiment to directly address the potential role of saccades, blinks and gaze position. First we investigated whether there was, in fact, any change in blink and/or saccade frequency around the time of a perceptual switch. This analysis focused on the two *Ignore* conditions; the conditions that gave rise to our most robust results. The analysis reveals that saccades occurred at an average rate of just under two per second in both conditions (black curves in *Appendix 1—figure 3B*, top plots), with a systematic change in saccade rate around the time of the perceptual switch. This change—first a drop, then a compensatory increase—resembles patterns observed in previous studies (*van Dam and van Ee, 2005*; *Einhäuser et al., 2008*). Blink rate, likewise, changes around the time of a perceptual switch, in agreement with the same studies, but is overall much lower (about one blink every seven seconds; black curves in *Appendix 1—figure 3B*, bottom plots). The change in both saccade rate and blink rate in association with perceptual switches means that saccade and blink-related pupil signals might be importantly reflected in our main results, if they are not captured by their dedicated regressors in the general linear model. Of note, however, for saccade rate the switch-related drop and subsequent increase are both of comparable magnitude and are spaced very closely together in time. As a result average saccade rate shows barely any change near the perceptual switch when assessed at a coarser time scale, especially when evaluated across both conditions (orange bars in *Appendix 1—figure 3B*, top plots, which depict saccade rate within intervals of 1 s). Given that the pupil response patterns that underlie our conclusions occur in both conditions and on a coarse time scale relative to the observed saccade rate fluctuation (e.g. the switch-related constriction lasts about two seconds; its correlation with preceding percept duration about a second; main text *Figures 2* and *5*), it is unlikely that this rapid fluctuation can explain those response patterns. For blink rate, the situation is less clear based on this particular analysis, because there is a net reduction in blink rate near switches, even when assessed at a coarse time scale (orange bars in *Appendix 1—figure 3B*, bottom plots).

To further investigate the role of saccades, and also gaze position generally, in our results, we performed a separate control experiment. Five observers performed a variant of our *On-screen, ignore* condition, in which the inner part of the rivalry stimulus was removed and a binocular fixation point was placed at the stimulus center, instead. This manipulation, as intended, led to a strong reduction in eye movements, which also precluded an OKN-based approach to the identification of switches. We therefore evaluated pupil size as a function of time relative to the physical, on-screen,

switches. In spite of the fixation point and the resulting lack of OKN, switches were still associated with a pupil constriction comparable to the ones observed in our main experiment (*Appendix 1—figure 3C*). This further argues against a critical role of saccades, and gaze position generally, in our finding of a switch-related constriction.

When it comes to blinks, we re-analyzed our main dataset to further examine their role in our results. Similar to *de Hollander et al., 2018* we examined pupil signals surrounding only those switches that are not close to any blinks: we removed from our switch regressors all switch events that were within 1.5 s of a blink, and moved those events to nuisance regressors, instead. This approach rules out any influence of the switch-related change in blink rate shown in *Appendix 1—figure 3B*. Nevertheless, the results of this re-analysis closely resemble those of the main analysis, with a robust switch-related constriction in both *Ignore* conditions (*Appendix 1—figure 3D*, top and center plots) as well as a significant dependence on preceding percept duration (*Appendix 1—figure 3D*, bottom plots). This result further reduces the plausibility of an important role of blinks in our results.

We performed one final analysis to directly address the role of gaze position in our results (a role that is also addressed by the analysis of *Appendix 1—figure 3C*). Here we asked where observers' horizontal gaze position on the screen was systematically different in time windows close to a perceptual switch as compared to at other times. Given our stimulus radius of 3.9 dva, OKN-related changes in horizontal gaze position that are confined to the stimulus area could cause apparent pupil size changes of up to a few tenths of a mm when using a video-based eye tracker (*Gagl et al., 2011*). That is not negligible compared to the magnitude of physiological pupil size changes previously observed in association with cognitive engagement (*McDougal and Gamlin, 2008*) or isoluminant stimulus changes (*Barbur et al., 1992*; *Young et al., 1993*), so an impact of this type of artifact cannot be ruled out on those grounds. On the other hand, an explanation centered on such artefacts would require a systematic tendency for participants to direct their gaze within a distinct eccentricity range during one or two seconds following a perceptual switch as compared to other time periods (i.e. during the seconds when we observed altered pupil size; main text *Figures 2*, *4*, *5* and *6*), while in reality gaze angle changed continually and multiple saccades happened each second (*Appendix 1—figure 3B*). To directly evaluate any tendency to look elsewhere during time periods near perceptual switches, we collected all gaze samples that fell within a 200 ms bin centered on any perceptual switch, as well as gaze samples that fell within other 200 ms time bins relative to the switch moments. The x-axes of the plots in *Appendix 1—figure 3E* indicate which time point is at the center of a bin in question, so an x-axis value of 0 corresponds to the bin that is centered on the switch itself and positive x-values indicate bins centered shortly after the switch. For each bin, we determined the distribution of horizontal gaze positions across all samples, and the widths of all those distributions are plotted along the y-axes of *Appendix 1—figure 3E*. The results suggest some tendency in the *On-screen* conditions for observers to direct their gaze relatively close to the screen center shortly after a perceptual switch, as compared to at other times. This tendency is highlighted by the inset of *Appendix 1—figure 3E*, which shows a relatively narrow distribution of horizontal gaze positions within the bin (gray) as compared to elsewhere (red; pink is the area of overlap). In spite of this apparent tendency, these results argue against an important role of gaze position in our main results, because the period of apparently altered gaze position lasts for only about half a second, and is not observed at all in the *Rivalry* conditions. The key pupil response patterns we report, in contrast, last longer and are consistent between the *Rivalry* conditions and the *On-screen* conditions (main text *Figures 2*, *4*, *5* and *6*).

In sum, a critical impact of saccades, blinks, or gaze position on our results is unlikely when considering the combination of (1) a marked consistency between key findings in our study and published findings, (2) various aspects of our main analysis approach that minimize the impact of such events, (3) no relevant change in saccade rate surrounding switches, (4) the results of a control experiment that minimized OKN, (5) the results of a control analysis designed to eliminate any role of blinks, and (6) the results of a control analysis designed to assess the role of gaze direction.

## Limitations of the OKN-based algorithm

We centered most of our analyses on switch events as identified by our OKN-based algorithm, because manual report and on-screen changes are not available as markers of switches in all

conditions. However, the OKN-based algorithm does not track perceptual switches perfectly: it over-estimates switch rates for slow switchers and underestimates it for fast switchers (main text *Figure 1C*), and it picks up on less than 100% of report-based switches and screen-based switches (main text *Figure 1D*). It is conceivable that the switches missed by this algorithm differ in some relevant way from the switches that it does detect, and that this systematically influences our estimate of the switch-related pupil response. To address that possibility we first performed an analysis to test the hypothesis the OKN-based algorithm selectively fails to detect switches when they are closely spaced, as may be suggested by the algorithm's tendency to underestimate switch rates for fast switchers. For this analysis we examined each reported perceptual switch in the *Rivalry Report* condition, and scored whether the OKN-based algorithm identified a switch during the 1.5 s interval preceding the key press report (an interval duration selected on the basis of main text *Figure 1E*). For each observer, we then sorted the reported switches by the percept durations that immediately preceded them (i.e. by the interval separating each reported switch from the preceding reported switch), and we also sorted the reported switches by the percept durations that immediately followed (i.e. by the interval separating each reported switch from the subsequent reported switch). Under the hypothesis that the OKN-based algorithm specifically tends to miss perceptual switches when they are closely spaced, one would expect OKN-based switches to be absent more often for reported switches for which these flanking percept durations are short. But the result of the analysis, shown in *Appendix 1—figure 4*, does not support that hypothesis: an OKN-based switch is observed shortly ahead of about 75% of all reported switches, and this percentage does not increase as a function of preceding or subsequent percept duration (in fact, it shows a modest decrease). This argues against the idea that the OKN-based approach selectively overlooks closely spaced perceptual switches.

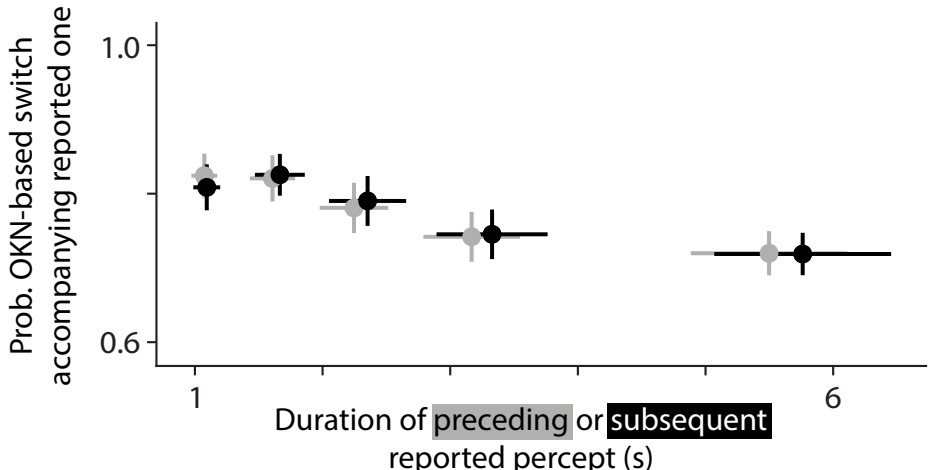

**Appendix 1—figure 4.** How performance of the OKN-based switch detection algorithm depends on flanking percept durations. Here, we investigated whether the reported percept duration that precedes (gray) or follows (black) a given reported switch, affects the probability that that reported switch is accompanied by a switch marked by the OKN-based algorithm. Across five percentiles of these flanking percept durations the probability is relatively stable at about 75%, and it decreases slightly for key press reports that terminate, or initiate, relatively long percepts.

Next, we analyzed whether the switch-related pupil response estimates that we obtained using our OKN-based algorithm, differed in any important way from estimates obtained using either key presses or on-screen events to mark perceptual switches. Those latter two markers are not available in all conditions, but they are available in some, which allows that comparison within those specific conditions. The results, shown in *Appendix 1—figure 5*, indicate that results are highly similar across the OKN-based approach (orange), the report-based approach (green) and the approach based on on-screen switches (yellow). The main difference is a temporal shift between the curves associated with the three approaches, entirely in line with the fact that the three event types are shifted in time relative to each other (main text *Figure 1D–E*). These results argue against the idea that our main

findings are importantly biased by the fact that our OKN-based algorithm is imperfect, and instead indicate that the main characteristics of the switch-based pupil response, as identified in our study, are consistently observed across all three approaches to identifying switches.

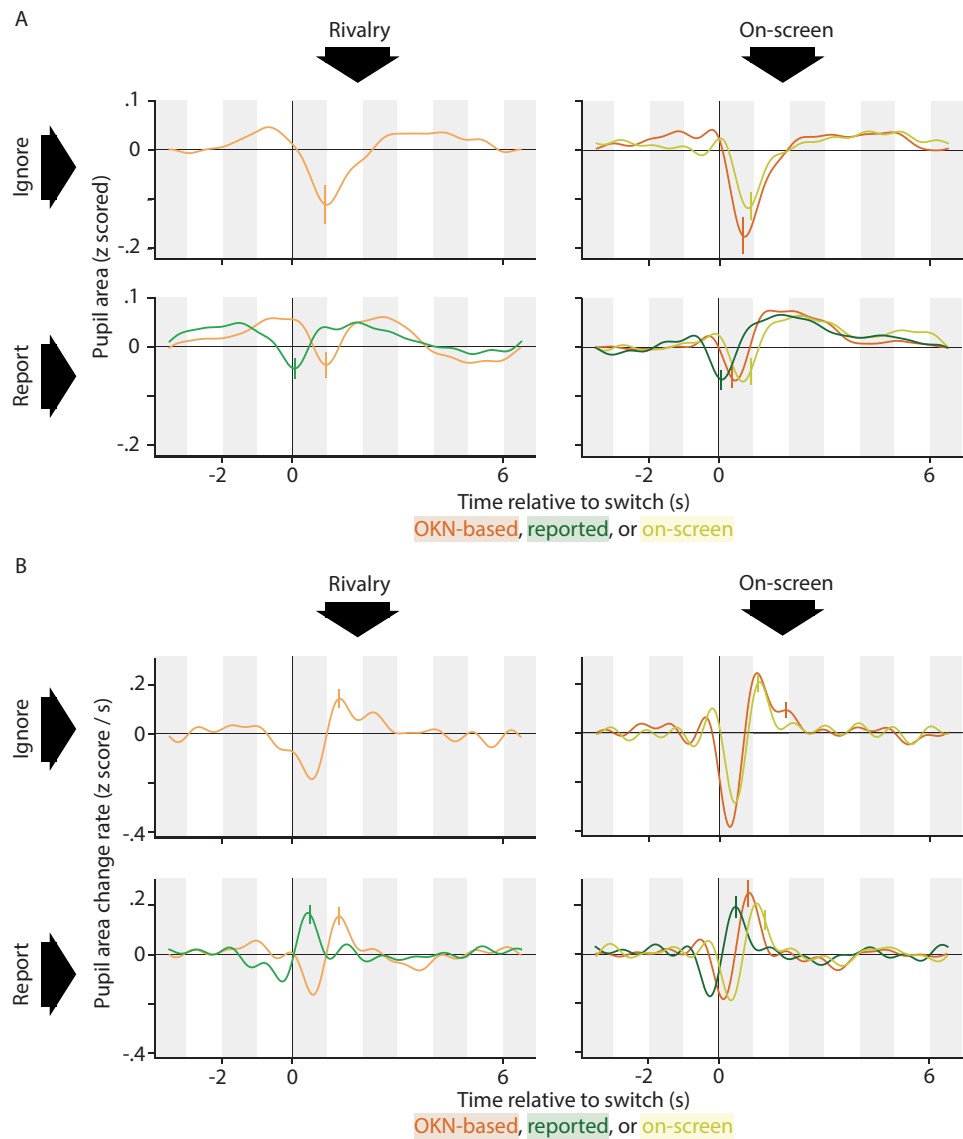

**Appendix 1—figure 5.** Switch-related pupil responses (both pupil area, top, and pupil area change rate, bottom) estimated while using three different methods for identifying perceptual switches. Although the OKN-based method is the only one that is available in all conditions, the match with the other two methods (based on manual report, and based on on-screen events) is good in those conditions where those other methods apply. The only marked difference between methods is a temporal shift in the estimated pupil response curves, in agreement with the fact that on-screen switches happen first, followed a little later by a reversal in OKN direction, and yet later by a key press response.

## Per-observer pupil response curves

The figures in the main text only show across-observer averaged pupil response curves (accompanied by some per-observer summary data in *Figures 3* and *4C*). To better convey the degree of variability or consistency in pupil response curves across observers, *Appendix 1—figure 6* shows all per-observer pupil response curves underlying the average plots of *Figure 2A*. Each panel

corresponds to an individual observer, and the four curves within each panel correspond to the four conditions.

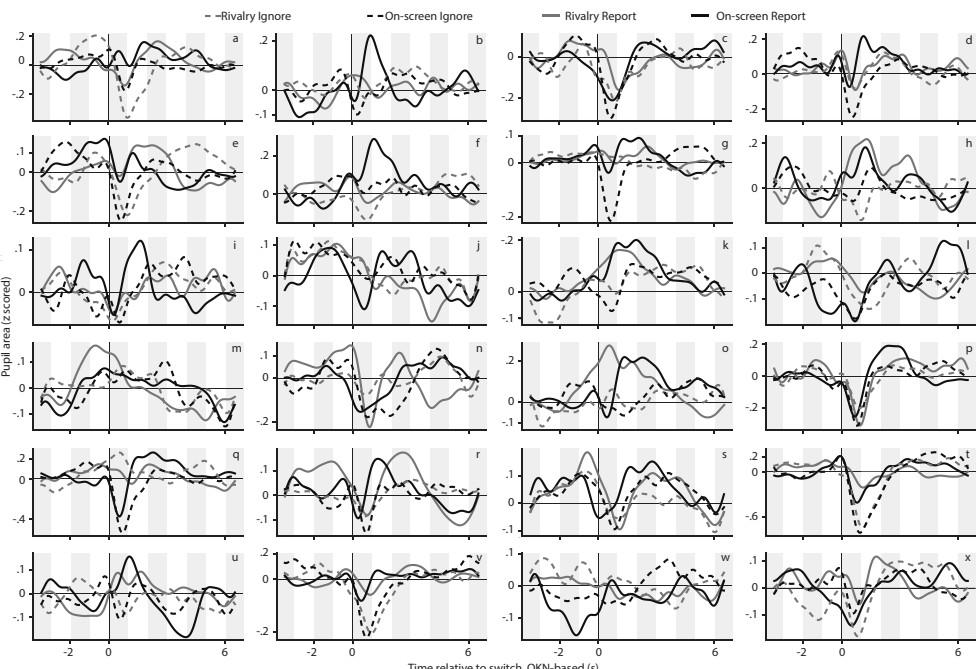

**Appendix 1—figure 6.** The per-observer pupil response curves that underlie the across-observer average curves of main text *Figure 2A*. Each panel corresponds to one of the included observers, and the four curves within each panel correspond to the four conditions (see legend on top).

Visual inspection of the curves reveals several noteworthy data patterns. First, the timing and shape of pupil response curves varies substantially between observers, consistent with existing reports of individual differences in pupil dynamics (e.g. *Denison et al., 2020*). Second, some participants (e.g. observers *j*, *m*, and *w*) do not show very clear pupil responses and, consequently, do not contribute much to the across-observer averages shown in the main text. This could be a matter of data quality rather than actual pupil motility: the ability of our deconvolution analysis to extract clean pupil responses depends on several factors, including an individual's perceptual switch rate and proportion of perceptual mixtures (see also *Appendix 1—figure 9*), as well as the frequency of nuisance events such as blinks. More importantly, in spite of the between-observer variability and the presence of some 'poor' observers, the figure reveals several data patterns that are consistent across a majority of observers. Many observers show a rapid constriction and re-dilation during the second or two immediately following the switch (e.g. observers *e*, *t*, and *v*), and during this short dip pupil size usually reaches a lower level in the *Ignore* conditions (dashed curves) than in the corresponding *Report* conditions (solid curves of the matching shade). Similarly, for many observers the pupil is temporarily dilated between about 1–3 s following the switch in the *Report* conditions, but not (or less so) in the corresponding *Ignore* conditions (e.g. observers *g*, *o*, and *r*). These patterns match those identified in association with the average curves shown in main text *Figure 2A*, and they are consistent with the notion that perceptual switches are accompanied by a transient pupil constriction that, if the switches are to be reported, party overlaps with a slightly later report-related dilation. An additional observation here, which cannot be made on the basis of across-observer averages, is that observers differ in which of the two response components (switch-related constriction or report-related dilation) predominates the overall pupil response. For some observers (e.g. *h* and *k*, and also *b* and *f* for the *On-screen* conditions) the response-related dilation is more pronounced, so that the net pupil response in the *Report* conditions (in which the constriction and dilation are superimposed) is almost a pure dilation. For others (e.g. *c*, *t*, and *v*) the switch-related constriction contributes much more to the overall response, so that the net response in the *Report* conditions is either a low-amplitude biphasic pattern or even simply a constriction, albeit one of smaller amplitude than in the *Ignore* conditions. This across-observer variety of net pupil responses in the *Report* conditions,

although surprising on the surface, is parsimoniously explained by the notion, favored here, that these net pupil responses are a combined product of two partially superimposed responses with opposite signs.

## Artefacts from overlapping pupil responses

In *Figures 5* and *6* of the main text we show three significant correlations between the switch-related pupil response and the duration of the preceding (two correlations) or subsequent (one correlation) percept duration. Because the pupil response spans across a few seconds it is conceivable that this type of association with flanking percept duration arises artefactually due to an incomplete separation, in our analysis, of overlapping responses to consecutive perceptual switches. We investigated this possibility by looking for data patterns that would be expected if this explanation was correct. In particular, under this explanation we would expect correlations as shown in main text *Figures 5* and *6* to differ systematically between observers with different average perceptual dominance durations. For instance, the observed group-level correlations with percept duration might stem primarily from observers whose switches are, on average, sufficiently closely spaced (by less than, say, 2 or 3 s) for overlap between adjacent switch-related pupil responses to be substantial. In addition, one might expect an observer's average percept duration to predict which exact section of the switch-related pupil response appears to depend on preceding, or subsequent, percept duration: for those observers whose preceding, or subsequent, switches occur relatively earlier this would be an earlier section. To evaluate whether any of these predictions are borne out in our data, we performed the analysis summarized in *Appendix 1—figure 7*. Here, we examined the same curves that we previously averaged across all observers to produce the mean curves of main text *Figures 5B* and *6B*, that is the curves that show the correlation between pupil area change rate and percept duration as a function of time during the pupil response period. However, this time we did not average those curves across all observers. Instead, we sorted the observers by average percept duration, and moved a sliding window across these sorted observers, averaging the per-observer curves of five observers at a time. In other words, we produced one average correlation curve for the five fastest switchers, another average correlation curve for the second through sixth fastest switchers, etcetera, down to the five slowest switchers. In the plots of *Appendix 1—figure 7* the gray levels along each horizontal slice can be thought of as representing one such average curve, and the percept durations (averaged across groups of five observers at a time) that correspond to the slices are plotted along the y-axis, with fast switchers at the top and slow switchers at the bottom. Each panel corresponds to one of the three panels of *Figures 5* and *6* that showed a significant correlation across all observers, and the time period of this significant correlation is indicated by dashed lines in the corresponding panel of *Appendix 1—figure 7*.

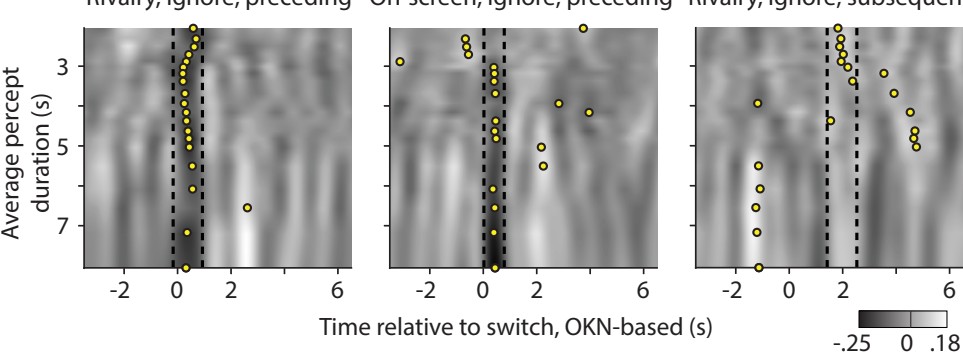

**Appendix 1—figure 7.** Re-analysis of the significant correlations between pupil response, and preceding (left and center plot) or subsequent (right plot) dominance duration, now separated out as a function of the observer's average percept duration. Gray values quantify correlations between pupil change rate (z-scored area per second) and preceding or subsequent percept duration (z-scored); correlations that are also shown, but averaged across all observers, in main text *Figures 5* and

*Appendix 1—figure 7 continued on next page*

*Appendix 1—figure 7 continued*

*6*. Here the correlations are averaged per groups of five observers, sorted by their average percept duration along the y-axes. Dashed lines delineate the time periods within which the average across all observers is significantly different from zero (periods that are marked by black bars in main text *Figures 5* and *6*). For each group of five, a yellow disk indicates the middle of the time period at which the five-person average reaches its most extreme value, after smoothing this five-person average by averaging within a sliding window as wide as the interval delineated by the dashed lines.

Both effects of preceding dominance duration (*Appendix 1—figure 7*, left and center panel) appear largely invariant with observers' average dominance durations: a vertical dark band in the relevant time period extends across nearly the full range (of about 6 s) of average dominance durations in both panels. One qualification here is that in the *On-screen Ignore* condition (center panel) the effect does not seem present in those observers with the very shortest dominance durations. These impressions are confirmed by identifying, for each 5-observer average curve, the moment at with the curve reaches its most extreme value (here we included both positive and negative extremes, within sliding windows as wide as the time window that shows the across-observer significant effect that is indicated by the dashed lines). The yellow dots in *Appendix 1—figure 7* show these moments. For the *Rivalry Ignore* condition (left panel), this extreme is found at a similar time across nearly all average percept durations. For the *On-screen Ignore* condition (center panel), this extreme is also found at a similar time in most cases, with most exceptions corresponding to the fastest switchers. Together these analyses indicate that the effect of preceding dominance duration is robust in our data, and unlikely to be an artifact caused by incomplete separation of overlapping pupil responses, which would primarily affect the data of the fastest switchers.

The right panel of *Appendix 1—figure 7* shows the same, but now pertaining to the correlation with subsequent dominance duration that was observed in the *Rivalry Ignore* condition (i.e. corresponding to main text *Figure 6B*, top left plot). Here, there is less consistency across average dominance durations, and an impression that the effect is primarily carried by the observers who have the shortest average dominance durations, of about 2–3.5 s. This might indicate a role of incomplete separation of overlapping pupil responses, especially given that the significant correlation with subsequent dominance duration was not observed until about 2 s after the switch event (main text *Figure 6B*)—close to the moment at which the next switch would occur, on average, for the observers who seem to carry this effect.

## Effect of preceding duration on response-related dilation

The main text shows a significant correlation between preceding percept duration and pupil response in the *Ignore* conditions where, arguably, the switch-related constriction is measured in isolation. It does not show a significant correlation in the *Report* condition where this constriction overlaps with the report-related dilation. One conceivable explanation is that the report-related dilation depends on preceding percept duration in a way that counteracts the dependence of the switch-related constriction. The direction of effect required for this would be one in which report-related dilations are larger following longer percept durations. That direction of effect would not be consistent with existing literature which, if anything, suggests that dilations might be larger following *shorter* percept durations, possibly related to the fact that switches that terminate shorter dominance periods are more surprising (*Kloosterman et al., 2015b*; *de Hollander et al., 2018*). Based on our design, it is difficult to obtain a clean estimate of the relation between preceding duration and the dilation that accompanies the switch report, because this dilation always overlaps with the switch-related constriction which, as shown, itself depends on preceding percept duration. Our design does, however, allow this type of estimate for the dilation response that accompanies the report of a so-called dot size probe. In the *Ignore* conditions, observers reported these transient stimulus changes, which occurred randomly and independently of perceptual switches (drawn from a uniform distribution between 3 and 8 s). *Appendix 1—figure 8*, top and center panel, show that these reports are associated with a pronounced dilation that starts shortly before the key press, in agreement with numerous reports of transient dilations associated with task execution (e.g. *Richer and Beatty, 1987*; *Hupé et al., 2009*; *Gilzenrat et al., 2010*), and also with main text *Figure 4*, bottom panels. *Appendix 1—figure 8*, bottom panel, shows how the amplitude of this

dilation relates to the duration that separates this probe report from the previous one. The panel shows no significant correlation, although a brief period immediately after the key press is marked by a non-significant negative correlation. The direction of this effect means that the report-related dilation is numerically larger following briefer intervals. Although not significant, this trend is consistent with existing observations (*Kloosterman et al., 2015b*; *de Hollander et al., 2018*) and also with interpretations in terms of surprise: when drawing from a uniform distribution with a limited range the instantaneous probability of a new event (given that it has not happened yet) increases monotonically with time passed since the previous event. The trend is opposite, however, to what would be needed to support the idea that, in our *Report* conditions, the switch-related constriction component and the response-related dilation component cancel each other out in terms of dependence on preceding percept duration.

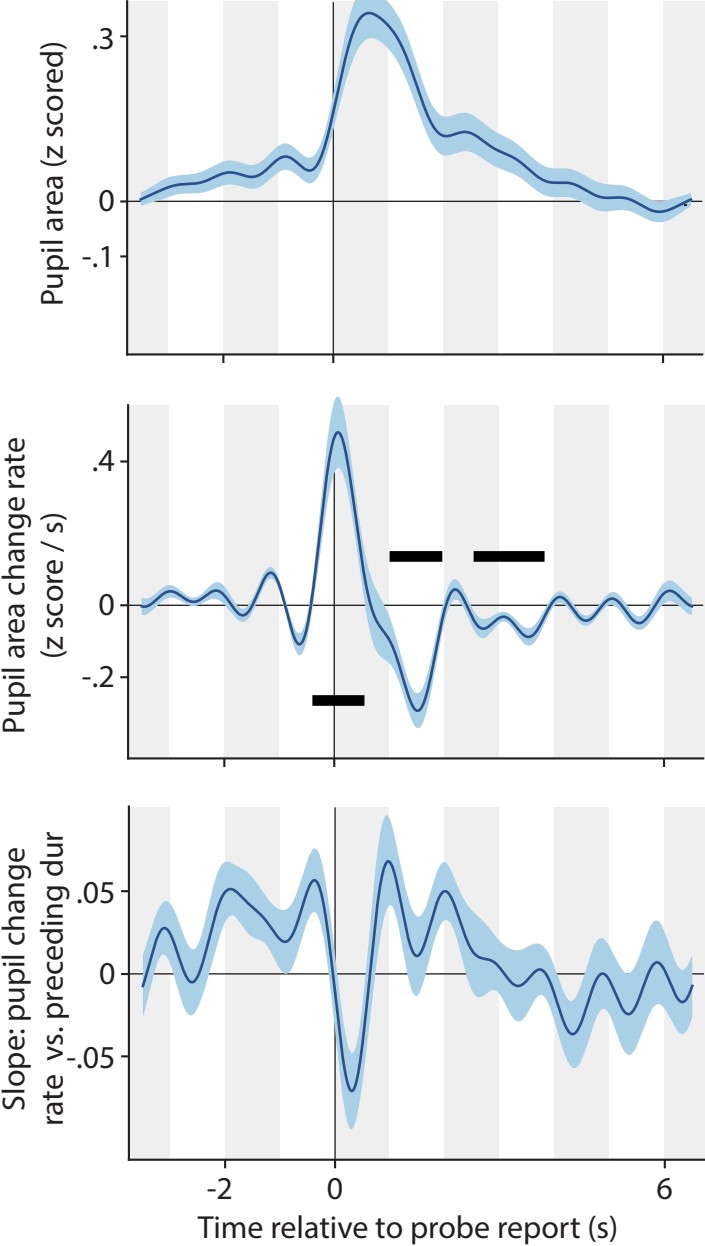

**Appendix 1—figure 8.** Pupil response associated with key presses in response to dot size probes in
*Appendix 1—figure 8 continued on next page*

*Appendix 1—figure 8 continued*

the *Ignore* conditions (both conditions combined). Pupil size (top panel) and pupil change rate (center panel) around the time of key press events reporting a dot size probe. Bottom: correlation between pupil change rate (z-scored area per second) and the amount of time passed since the previous key press (z-scored). All plotting conventions are the same as those introduced in main text *Figures 2*, *4*, *5* and *6*.

## Perceptual mixtures

In the *Report* conditions, observers could indicate experiencing either of the two exclusive percepts, or a mixture made up of parts of the two monocular displays perceived at the same time. Although there is no single clear criterion as to what counts as 'too much' mixture perception for our purposes, it is clear that observers who experience more mixtures are less suitable, as our study focuses on perceptual switches between the two exclusive percepts. We quantified mixture perception in three different ways and decided to discard the data of two participants who consistently scored the highest of all participants on each of the three measures. The three panels of *Appendix 1—figure 9* illustrate the three approaches, with each dot marking one participant and the two red dots in each panel marking the two discarded participants. The top panel shows the proportion of viewing time spent experiencing a mixture percept (as based on key press reports); a standard measure of mixture perception. The two participants marked in red are outliers on this scale, and are the only observers who reported experiencing a mixture more than half of the time. But one could argue that the validity of our pupil analyses is dependent, not so much on what proportion of time is occupied by mixtures, but on how long mixtures last while transitioning from one exclusive percept to the other. As long as such transition durations are brief the pupil responses we quantify can reasonably be interpreted as associated with perceptual switches. The center panel, therefore, shows the average duration of mixture periods for all observers. Again, the same two observers score the highest. Finally, one could argue that the average mixture duration is not critical—an observer might experience a small number of excessively long mixture periods that pull up this average—as long as there is a sufficient number of brief transitions that contribute heavily to our switch-related pupil estimates. The bottom plot, therefore, shows for each observer the proportion of all mixture periods that last longer than 1 s. Again, the same two participants score highest: they are among only four observer for whom over half of the mixture periods last longer than a second. Because those two participants scored highest on all three measures, and also because the order of the next-highest scoring participants was inconsistent across the three measures, we decided to base our main analyses on the data from all observers except those two.

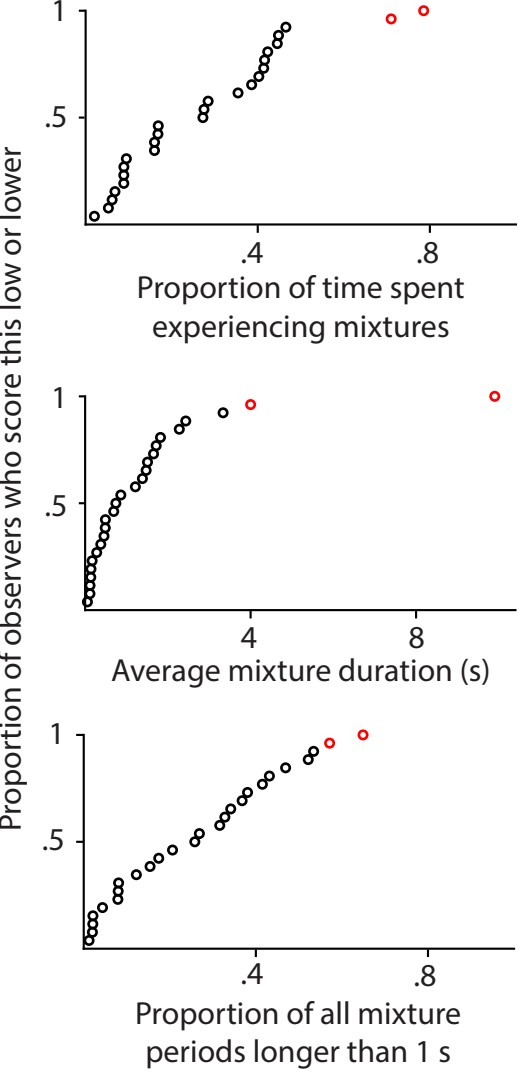

**Appendix 1—figure 9.** Perceptual mixtures: percepts that feature parts of both eyes' images. Each plot uses a different measure to quantify the reported prevalence of perceptual mixtures for individual observers. Each disk corresponds to one observer. Data from the two observers that are indicated in red here were not included in the analyses for any of the other figures in this paper, because those observers scored high on all three indices of the prevalence of perceptual mixtures.

