## [Decision Letter]

**Acceptance summary:**

Pupillometry is an increasingly accessible tool for the non-invasive readout of brain activity. This study used pupillometry to explore the neural processing underlying perception and dissociate them from action-related neural processing. Results reveal changes in pupil size that are reliably different depending on the task. Such approaches can be very useful in deciphering which of the myriad factors that can affect pupil size are active under specific, controlled conditions.

**Decision letter after peer review:**

Thank you for submitting your article "Separable pupillary signatures of perception and action during perceptual multistability" for consideration by *eLife*. Your article has been reviewed by 3 peer reviewers, and the evaluation has been overseen by Miiram Spering as Reviewing Editor and Floris de Lange as the Senior Editor. The following individuals involved in review of your submission have agreed to reveal their identity: Wolfgang Einhaeuser (Reviewer #1); Alexander Pastukhov (Reviewer #3).

Essential Revisions:

The reviewers and reviewing editor agree that this study is well-conceived and executed. However, there are some questions and concerns about the analyses and conclusions based on the results shown.

1) Method / Luminance issues

The following issue was brought up during the discussion amongst all reviewers and the reviewing editor and is potentially of concern. At minimum, the reviewers would like to see clarification in the manuscript.

It appears that there might be a methodological concern. Changes to the average luminance emitted from the screen differ between the ignore and the rivalry condition. The methods (p.15) say that the dark and the light dots are not evenly balance around the mean luminance of the background screen (l.551 following): "..background (34.5 cd/m2). Half of the dots of a given color were lighter than the background (62.8 cd/m2) and half were darker (19.0 cd/m2)." Assuming equal numbers of dots, the average luminance of the dots would be (62.8 cd/m2 + 19.0 cd/m2) / 2 = 40.9 cd/m2. This contrasts with the 34.5 cd/m2 for the background to the dots, they report. In the "ignore" condition, there is a dot size probe that shrinks and grows the dots back, which subjects have to report. Because of this imbalance, the mean luminance of the screen will change slightly. It is small but might still potentially affect pupil size in this essential control condition. So, as they shrink the dots and more space is taken over by the background, the stimulus gets overall darker. This might unleash complex changes in pupil size. This is further complicated by the use of a staircase procedure for different dot size changes on different trials for individual subjects.

One approach to clarifying this could be to look at the relationship between the time of OKN detected switch and the time at which the mini-local-looming effect (shrink and grow dots) is presented. If these are quite separated in time, then the associated luminance changes are less of a concern, since the switch related pupil changes will be separated in time from the potential luminance change related changes. But, if there are a significant number of trials for which this luminance change occurs close to the OKN detected switch, then this could be a serious concern. To summarize: it seems important to clarify what the actual measured change in luminance was due to this shrink-and-grow manipulation? And does it produce a change in pupil size?

Moreover, the pupil response is consensual, so if there are luminance inhomogeneities in the monitor and the luminance presented at the two eyes is different, that will drive changes in pupil size that are unrelated to perceptual stability. With binocular stimulation, the stability at each screen location has to be confirmed. If the authors could show that the luminance of the stimulus presented to each eye was constant over time, this would allay this specific concern.

Finally, it would be good to double check that the constriction does not depend on the polarity of the switch (magenta to cyan vs. cyan to magenta). If one stimulus is perceived brighter than the other, the constriction when going from the darker to the brighter stimulus would possibly not be compensated by the dilation when going from the brighter to the darker stimulus (due to distinct time courses), which would emphasize the constriction effect over other stimuli. I do not expect this to be a main component, but it would be good to check as it might contribute to the effect that the constriction seems stronger in this stimulus than in the examples from the literature.

2) OKN analysis and switch rate

The reviewers are asking for additional detail to ensure that the elicited OKN response is sufficiently reliable and does not bias the results in any way. More specifically: when using similar stimuli as the authors, it has been observed that participants follow a single dot rather than the full pattern, especially if the dots are of unlimited lifetime. In addition to saccade rate and blink rate, it would be interesting to report the eye position relative to the time of the switch. This would also provide an additional argument against any artefact of eye-position effects on apparent pupil size. Curiously, the authors call the OKN a "pursuit" eye movement (l.113), which seems to hint at the notion that they sometimes indeed observe pursuit (of a single dot) rather than OKN. For additional clarity, unless pursuit is indeed the dominant response, the word "pursuit" in line 113 should be deleted.

A potential methodologically concern is the under and overestimation of mean switch rate via OKN (figure 1C). OKN estimates are all within.4-.8 range, whereas for self-report rates differ from 0.2 to over 1. Further analysis would be helpful. E.g.., there is a group of participants with ~0.2 switches per second reported mean switch rate but the OKN estimates (0.4-0.6 switches per second) would suggest that these participants did not report roughly half of perceptual switches. I think it would be helpful if the authors elaborated on what kind of switches went unreported (or, conversely, what kind of events led to false alarms): switches before very short dominance phases (could be too fast to report via key presses), to return transitions, etc.

3) Graphical presentation of data

All reviewers had a hard time following some of the figures, especially Figure 2. Below is a list of points to address, some specific to Figure 2, others more general. Perhaps most importantly: figures in this manuscript do not convey information clearly. Showing data from an example/representative subject and then distributions (or boxplots or scatter plots) of the relevant parameters could bring out the main effects in a visually convincing way. It seems a bit unusual to show only population averaged time-series as result plots (as all main figures are in this manuscript). Please consider this suggestion carefully.

1. Figure 2 is too dense. The reviewers suggest to split this figure (and figure 3) into separate figures for pupil area and change so the reader can directly compare graphs.

2. Additionally, there are some issues with the labeling in Figure 2. Minimally, it would be helpful to label constriction and de-dilation episodes that are referred to in the text for all four conditions so text in lines 190-192 can be matched with plots: "The fact that only the Report conditions show the re-dilation going past baseline and being followed by a final constriction".

3. Figure 2. It would be also useful to have same plots centered at the moment of an actual change for on-screen condition.

4. Figure 2. There is plenty of space for the central figure, so no point in abbreviating condition into "Ign.", "On-sc.", etc.

5. For the average traces of figure 2, how do the authors deal with the fact that a single datapoint can contribute at multiple time points, and this effect is stronger for fast switchers than for slow switchers? Some of the previous studies have cut the data half-way to the next switch to avoid this issue, although this is not completely free of issues, either. If no such step is done, this could shift the mean of the average curve relative to 0 (the overall mean is of course zero due to z-scoring, more data points close to switches contribute more likely multiple times), which does not seem to be the case here (but I do not see a switch-wise baseline-correction either, which would in itself be problematic, as then the constriction could be in fact a rebound from a previous dilation). Maybe the authors could add some lines on the averaging procedure, just to make sure that these issues are not missed. I understand that the section "Artefacts from overlapping pupil responses" addresses a similar issue for the later analyses, but a direct comment on this for figure 2 would be helpful (and maybe a replotting of figure 2 with the half-way cut in the Appendix?).

6. Figure 2. The amplitude and time-courses of the positive rate of change of pupil following constriction appear to be comparable in the Ignore and Report conditions. I might be missing something here, but I'm not clear how this is consistent with different pupil dilations around these epochs (i.e. when the pupil rate of change is positive). That is: the same positive rate of change over the same period of time should lead to the same amplitude of dilation.

7. Figure 2 (and following figures). In addition to the pupil size and rate of change, did the authors try and use the time to peak or maximum rate of change (dilation or constriction) as a measure to compare between conditions?

8. Figure 2. For the epoch preceding a switch, the form of the pupil- and rate of change of pupil-timeseries appear quite different to me, in both the Rivalry conditions. That is, in the 2 panels in the left column of figure 2, the Report condition seems to be associated with a sustained dilation before and beyond t=0 while in the Ignore condition, the pupil starts constricting before t=0. Is this difference really not reliable (significant in some way) across subjects?

4) Result clarification

The reviewers raised a number of specific concerns about the interpretation of results. Some of these issues (those related to the figures) are probably easily resolvable, others might need more clarification and rewriting of parts of the manuscript.

1. Page 6, Line 198. "This pattern of…". Could the authors clarify how this claim is supported? The dilation is significant when the pupil data are aligned with the switch and with the response, ie – with the perceptual effect and the response effort. That is, the significant dilation could be associated with either or both: perceptual effect and response effort. I don't see how the result shown here alone can support this claim. I agree with the result supporting the claim that dilation but not constriction is related with response effort, but that isn't too surprising since the former precedes the latter.

2. Figure 3. The paired t-test shows a reliable difference between the rivalry and on-screen conditions at the end of the constriction phase and through the early part of the dilation phase (as early as about 1 sec). To me, this suggests a significant contribution of dilation related neural activity when comparing rivalrous and explicit changes in the stimulus. Isn't this contrary to the main claim that pupil-linked arousal (presumably involving noradrenergic mechanisms, among others) is not related to spontaneous switching? Alternatively, if I'm misreading the numbers and position of the black line, so that it does not cover the dilation phase, this is another reason to show the data another way. For example, plot the distribution of pupil rate of change for the significant epoch so that one can decipher whether it is a dilation or constriction phase.

3. Figure 4B. It's likely that this is due to using the mean+-sem to display the data, but when I look at the top and bottom panels of this figure, it appears that there is a large dilation on the bottom left compared with top left (Report and Ignore for Rivalry switches respectively) and on the top right compared with bottom right (Ignore and Report for On-Screen switches, respectively). Since these are shown as not-significant from the ANOVA analysis, can the authors reconcile why there appears to be a relatively large, nonzero mean+-sem range for dilation (at about 2-4 sec)?

4. Figure 4B. Similar to point 4 above, the bar indicating significance appears to bridge the inflexion point for the pupil derivative – suggesting that the late constriction and early dilation phases, following the switch, are reliably related to the switch. Again – this is contrary to the main claim that arousal (or noradrenaline) linked dilation is not associated with the switch. Could the authors clarify this?

5) Pupil pathways / result interpretation

The reviewers would like to encourage the authors to try and include more of what we do know about neuromodulation and the cortical control of pupil pathways to frame the hypothesis and interpret the results.

The discussion leading to the conclusion about the relative roles of parasympathetic and sympathetic systems is somewhat weak. Since this is claimed as a main finding, it needs work both to improve the accuracy and strength of the argument. For example, line 409: "… cortical inhibition of the Edinger-Westphal nucleus,.…": there is no direct cortical inhibition of the EWN (or any region). Previous work has shown both: pupil dilation and constriction resulting from cortical (frontal eye fields, FEF) microstimulation (Lehmann and Corneil, 2016; Ebitz and Moore, 2017). One pathway for such effects is FEF projections to the pretectal olivary nucleus that then modulates activity in the EWN. Another pathway for cortical effects to reach the EWN is via the superior colliculus.

In the view of one of the reviewers, one convincing argument consistent with their findings (and that would allow them hypothesize about the putative pathways) could center on the idea of transient lapses in attention (or in pupil linked arousal). The Discussion starting on line 387 starts out in this direction. For example, transient reductions in the activity of the noradrenergic system could lead to reduced inhibition of EWN (that is effected via α-2 adrenergic receptors on EWN neurons). So, in addition to any cortical effects (possibly via the pretectal olivary nucleus) on the parasympathetic system, there could reductions in the activation of the sympathetic pathways, leading to disinhibition of the EWN. This would result in pupil constriction consistent with the results shown here.

Related to the above point, the Discussion (paragraph starting on line 368) cites previous work that found dilation linked with perceptual switching (or cognitive switching, in some of the references). However, in some of those studies, the dilation linked effects were derived from measures of baseline pupil size. Those studies predicated their analyses on this measure (baseline pupil), using it as an indicator of arousal (or attentional) state preceding action-selection, and they found reliable links between pupil linked brain state and successful detection of changes in stimulus (or stimulus reward associations, eg change-points in explore-exploit choice tasks). Do these data allow for a baseline contingent analysis?

[Editors' note: further revisions were suggested prior to acceptance, as described below.]

Thank you for resubmitting your work entitled "Separable pupillary signatures of perception and action during perceptual multistability" for further consideration by *eLife*. Your revised article has been evaluated by Floris de Lange (Senior Editor) and a Reviewing Editor.

The manuscript has been improved but there are some remaining issues that need to be addressed, as outlined below:

Essential revisions:

1) It is critical that the authors please revisit their statistical tests, and how they are reported. It is still unclear whether the reported statistics are calculated within or across subjects, and whether tests are done on combined data. If yes, the authors need to also report within-subject effects. Rev. 2 and the reviewing editor still have some doubt about the reliability of effects, and the authors need to pay close attention to this issue before resubmitting their revision.

2) Clarify and add to the description of OKN analysis, as described in detail by rev. 3 below.

*Reviewer #2:*

It is still unclear to me (after reading the Methods/Data Analysis section) whether the reported statistics are calculated within or across subjects. I appreciate the responses to questions in the initial review, but I think this needs more clarity. For example, the responses in "Result clarification": how are these tests done? It looks like they are done on the combined data. But are there within subject effects and if so, what proportion of subjects show the main effects that are highlighted in the Results section? This is an important point because if the Conclusion about the general applicability of these findings is to be, the findings needs to be reliable in at least a majority of subjects.

The expanded Discussion is an improvement. Another point that might be helpful to consider is that it is mostly the prefrontal and cingulate cortices that provide descending control of LC excitability. Since this study is centered on perceptual effects and not on cognitive or physical effort, the scope of the work is consistent with a sensory cortical dominant network so that cortical regulation of LC-linked arousal is not a factor.

Line 178: Rate of change was used in an earlier study to show LC spike linked changes in pupil size in Joshi et al., 2016 (their Figures 4-6 and 8).

*Reviewer #3:*

The authors address all my concerns. The only change I would suggest is using part of their response in the manuscript itself, as it conveys the idea very nicely. I would use it to replace / complement the second half of the last sentence in "Limitations of the OKN-based method of identifying switches". The part of the response is "Finally, although the OKN-based algorithm, then, is not perfect, for our purposes the critical requirement is that the OKN-based algorithm marks time points that are strongly associated with perceptual switches; not that it marks zero spurious time points nor that it marks a time point for every single perceptual switch. This is especially true because our key analyses concern comparisons between conditions while keeping the switch-identification method constant; not comparisons between identification methods."

---

## [Author Response]

Essential Revisions:The reviewers and reviewing editor agree that this study is well-conceived and executed. However, there are some questions and concerns about the analyses and conclusions based on the results shown.1) Method / Luminance issuesThe following issue was brought up during the discussion amongst all reviewers and the reviewing editor and is potentially of concern. At minimum, the reviewers would like to see clarification in the manuscript.It appears that there might be a methodological concern. Changes to the average luminance emitted from the screen differ between the ignore and the rivalry condition. The methods (p.15) say that the dark and the light dots are not evenly balance around the mean luminance of the background screen (l.551 following): "..background (34.5 cd/m2). Half of the dots of a given color were lighter than the background (62.8 cd/m2) and half were darker (19.0 cd/m2)." Assuming equal numbers of dots, the average luminance of the dots would be (62.8 cd/m2 + 19.0 cd/m2) / 2 = 40.9 cd/m2. This contrasts with the 34.5 cd/m2 for the background to the dots, they report. In the "ignore" condition, there is a dot size probe that shrinks and grows the dots back, which subjects have to report. Because of this imbalance, the mean luminance of the screen will change slightly. It is small but might still potentially affect pupil size in this essential control condition. So, as they shrink the dots and more space is taken over by the background, the stimulus gets overall darker. This might unleash complex changes in pupil size. This is further complicated by the use of a staircase procedure for different dot size changes on different trials for individual subjects.One approach to clarifying this could be to look at the relationship between the time of OKN detected switch and the time at which the mini-local-looming effect (shrink and grow dots) is presented. If these are quite separated in time, then the associated luminance changes are less of a concern, since the switch related pupil changes will be separated in time from the potential luminance change related changes. But, if there are a significant number of trials for which this luminance change occurs close to the OKN detected switch, then this could be a serious concern. To summarize: it seems important to clarify what the actual measured change in luminance was due to this shrink-and-grow manipulation? And does it produce a change in pupil size?

We agree that the looming manipulation associated with the task during the 'Ignore' condition involved a small but real change in net luminance. We would like to point out that our use of a general linear model framework for estimating switch-related pupil responses alleviates the concern that such a luminance change might systematically impact our estimate of the switch-related pupil response, to the extent that various components of pupil size change combine linearly (and can, therefore, be disentangled using a linear model), and to the extent that the moments at which the looming events occur are sufficiently independent from the moments at which the switches occur (otherwise the ability to disentangle is compromised). The former condition (the one about linearity) is at least approximately met, given the success of general linear model approaches to pupil analysis in our work as well as that of others (for instance Hoeks, B. and Levelt, W. J. M. (1993) Behav Res Methods Instruments Comput 25, 16–26; Gee, J. W. de, et al. (2014) PNAS 111, E618-25; Wierda, S. M., et al. (2012) Proc National Acad Sci 109, 8456–8460), but it is a distinct possibility that this former condition is not fully met (i.e. overlapping pupil response components might not combine perfectly linearly). The latter condition (the one about independence between switch event timing and looming event timing) is certainly fully met, at least in our 'On-screen Ignore' condition in which the timing of on-screen switches is predetermined (by the timing of reported switches in earlier 'Report' trials) and the moments of the looming events are drawn randomly by the experiment code. The fact that the data patterns that we base our conclusions on are highly similar across the 'On-screen' conditions and the 'Rivalry' conditions, thus, provides some reason to believe that the looming-related luminance changes play no critical role in our findings.

With all that being said, we agree that events that involve luminance changes deserve close scrutiny in any pupillometry study. We have now performed two further analyses to estimate the potential impact of the looming manipulation on our findings. First, we have now explicitly evaluated the pupil response to non-reported looming events, and we observe that this response is not significant at any time point surrounding the event. In the revised manuscript this is now detailed in Appendix 1-figure 1A and surrounding text. The non-reported looming events, in contrast to the reported ones, isolate the luminance-related pupil response, so this result suggests that the change in luminance involved here is sufficiently small as not to generate a notable pupil response. (Reported looming events, on the other hand, are associated with a classic task-related pupil dilation, as was reported in our original appendix; current Appendix 1 figure 7). As a second measure we have now performed a modified analysis of switch-related pupil responses in the 'Ignore' conditions, this time using only switches that are not close to a 'looming event' (separated from the closest looming event by at least 1.5 s on both sides). The results, detailed in Appendix 1-figure 1A and surrounding text in the revised manuscript, show that the switch-related pupil responses obtained in this modified analysis are highly similar to the ones obtained in the original analysis that did not separate switches into ones that were, and ones that were not, close to a looming event. This further argues against an important impact of the looming events, and associated luminance changes, on our findings.

Moreover, the pupil response is consensual, so if there are luminance inhomogeneities in the monitor and the luminance presented at the two eyes is different, that will drive changes in pupil size that are unrelated to perceptual stability. With binocular stimulation, the stability at each screen location has to be confirmed. If the authors could show that the luminance of the stimulus presented to each eye was constant over time, this would allay this specific concern.

We do not, unfortunately, have the technical ability to directly establish empirically the extent to which the luminance output of the screen was constant over time during our trials -- our photometer can measure luminance at discrete moments in time, but cannot measure luminance repeatedly across an extended time interval or across many distinct screen locations in an automated fashion, as would be required to assess this degree of constancy. However, the concern that unintended luminance changes in the stimulus, stemming from differences in light output between screen locations, would importantly affect our main findings regarding switch-related responses is much reduced by the fact that those main findings hold for both the 'rivalry' conditions and the 'on-screen' conditions. In particular, an explanation in terms of screen inhomogeneities would require switch events to systematically be related with moments at which dots happen to pass over screen locations with deviating light output. In the 'on-screen' conditions, however, the switch events occur at times that are predetermined by the key press events of earlier trials of the 'rivalry' kind (i.e. the switches are being 'replayed' according to the timing with which switches were reported during earlier rivalry trials), and the positioning of the dots is randomly determined by the computer. This effectively rules out a systematic relationship between switch event timing and any particular dot location in the 'on-screen' conditions. It is important, in this context, that the actual positions of the dots in the random dot field during the 'on-screen' trials were not replayed according to those earlier rivalry trials. Instead, actual dot positions were randomly drawn at the start of each trial. This means that, even in a scenario in which systematic variations in overall luminance coincided with switch events in the rivalry conditions (for instance because switches could be triggered by occasions where an above-average number of randomly positioned dots happen to move over a screen region where there is a relatively high light output because of screen inhomogeneities), such luminance variations would not be systematically associated with switches in the 'on-screen' trials. In our original manuscript we neglected to mention how dot locations related between conditions, and it would have been reasonable for a reader to assume that we 'replayed' dot locations between 'rivalry' trials and 'on-screen' trials, just like we replayed switch moments. We have now added the relevant information about dot locations to the manuscript (on page 21). In sum, although we cannot rule out that pupil size during our experiment may have, in part, displayed some variation related to incidental changes in stimulus luminance, for instance related to screen inhomogeneities, we do not see any scenario under which such variations can explain our main findings regarding switch-related pupil size changes because those generalize between the 'rivalry' and 'on-screen' conditions.

Finally, it would be good to double check that the constriction does not depend on the polarity of the switch (magenta to cyan vs. cyan to magenta). If one stimulus is perceived brighter than the other, the constriction when going from the darker to the brighter stimulus would possibly not be compensated by the dilation when going from the brighter to the darker stimulus (due to distinct time courses), which would emphasize the constriction effect over other stimuli. I do not expect this to be a main component, but it would be good to check as it might contribute to the effect that the constriction seems stronger in this stimulus than in the examples from the literature.

Yes, this is a good idea, especially because, as the review report rightly points out, light-related constrictions tend to be faster than light-related dilations so that the two would not completely average out in our design when we aggregate across switches in two opposite directions. We have now repeated our analysis of switch-related pupil responses in both 'Ignore' conditions, now performing the analysis separately for switches toward one color and for switches toward the other color. For the 'Rivalry Ignore' condition we have now furthermore performed the analysis separately for switches toward dominance associated with one monitor/eye and for switches toward dominance associated with the other monitor/eye. (In the 'On-screen Ignore' condition this latter analysis would make no sense because there stimuli were presented binocularly.) We did not perform additional analyses for the 'Report' conditions because our main results indicate that the pupil response in those conditions is a mixture of both the perception-related pupil response (to which this reviewer point applies and which is isolated in the 'Ignore' conditions) and the task-related pupil response (to which this reviewer point does not apply). The results of our re-analysis can be found in the revised manuscript in Appendix 1-figure 2 and surrounding text. The important finding is that the switch-related pupil response is comparable, and statistically indistinguishable, regardless of the direction of the switch in terms of color and eye/monitor.

2) OKN analysis and switch rateThe reviewers are asking for additional detail to ensure that the elicited OKN response is sufficiently reliable and does not bias the results in any way. More specifically: when using similar stimuli as the authors, it has been observed that participants follow a single dot rather than the full pattern, especially if the dots are of unlimited lifetime. In addition to saccade rate and blink rate, it would be interesting to report the eye position relative to the time of the switch. This would also provide an additional argument against any artefact of eye-position effects on apparent pupil size.

We have now performed an analysis of the relation between horizontal gaze position and perceptual switches. The results are presented in Appendix 1-figure 3E and surrounding text. The analysis suggests some tendency in the On-screen conditions for horizontal gaze position to be concentrated more closely near the center of the screen during the time period immediately surrounding perceptual switches than at other times. Still, two aspects of this result argue against the notion that this apparent tendency would provide a gaze-based explanation of any of the important data patterns reported in our main text. First, and most importantly, the same tendency was not present (not even qualitatively) in the Rivalry conditions, whereas all important data patterns we report generalize between the Rivalry and On-screen conditions (main text Figures 2-4). Second, in those conditions where the data did suggest a tendency for a more central gaze position surrounding switches, the time window during which the tendency was observed was too brief (under a second) to explain our observed switch-related pupil responses in terms of foreshortening artefacts (the significant part of the switch-related pupil responses lasts about two seconds; main text Figures 2 and 3).

Curiously, the authors call the OKN a "pursuit" eye movement (l.113), which seems to hint at the notion that they sometimes indeed observe pursuit (of a single dot) rather than OKN. For additional clarity, unless pursuit is indeed the dominant response, the word "pursuit" in line 113 should be deleted.

We agree. Sorry about that. We have no reason to suspect that observers' eyes were at any time following individual dots, and we have now removed the word 'pursuit' from the text.

A potential methodologically concern is the under and overestimation of mean switch rate via OKN (figure 1C). OKN estimates are all within.4-.8 range, whereas for self-report rates differ from 0.2 to over 1. Further analysis would be helpful. E.g.., there is a group of participants with ~0.2 switches per second reported mean switch rate but the OKN estimates (0.4-0.6 switches per second) would suggest that these participants did not report roughly half of perceptual switches. I think it would be helpful if the authors elaborated on what kind of switches went unreported (or, conversely, what kind of events led to false alarms): switches before very short dominance phases (could be too fast to report via key presses), to return transitions, etc.

We agree that the imperfect correlation between switch rate estimates from various methods is reason for close scrutiny. We have now performed an additional analysis along the lines suggested in the review comment, to better understand where the discrepancy comes from. Specifically, for each manually reported percept switch during the Rivalry Report condition we scored whether or not the OKN-based algorithm identified a switch in the time window where that would be expected (i.e. shortly before the manually reported switch because of reaction time delay), and we also recorded the percept durations that immediately preceded, and immediately followed, that manually reported switch (percept durations, as computed here, were based on the key presses; not the OKN-algorithm). The result is shown in Appendix 1-figure 4. To summarize the main finding: the probability of the OKN-based algorithm marking a switch in the expected time window was about 0.75, and this number depended only modestly on the flanking percept durations: the probability was a bit higher if either the preceding or the subsequent percept duration was short (slightly above 0.8 for durations below about 2 s) and a bit lower if either of those durations was long (slightly above 0.7 for durations around 5 or 6 s). This result provides no clear leads as to what characterizes the perceptual switches that are missed by the OKN algorithm, but it does argue against the specific idea that the algorithm selectively misses percepts of a particular duration.

In our Discussion section (page 19), we now elaborate on a tentative interpretation of the particular nature of the imperfect correlation between the results of the two methods. To summarize that interpretation: as shown in Figure 1, the across-observer correlations between manually reported switch rates and OKN-based switch rates shows that OKN-based method produces a relative overestimation of the switch numbers for slow switchers yet a relative underestimation of switch numbers for fast switches. For this reason we tentatively conclude that the OKN-based method marks a spurious switch at some fixed rate per unit time (i.e. every pupil time series of a given length in our experiment will erroneously trigger the algorithm with a set probability), and additionally fails to mark an actual switch at some fixed rate per actual switch (i.e. each pupil time series that accompanies a manually reported switch in our experiment has a non-100% probability of triggering the algorithm). For slow switchers the former outweighs the latter, resulting in a net overestimation of the number of switches, yet for fast switchers the latter outweighs the former, resulting in a net underestimation.

Finally, although the OKN-based algorithm, then, is not perfect, for our purposes the critical requirement is that the OKN-based algorithm marks time points that are strongly associated with perceptual switches; not that it marks zero spurious time points nor that it marks a time point for every single perceptual switch. This is especially true because our key analyses concern comparisons between conditions while keeping the switch-identification method constant; not comparisons between identification methods. We do understand the reviewers' interest in understanding exactly what type of switches are being missed by our algorithm: it is at least a theoretical possibility that those switches have a different pupillary signature than the ones we don't miss. But, as discussed, our examination of a relation with percept duration yielded a negative result, so at present we remain agnostic as to which types of switches are missed, and we see no specific reason why missed switches would have a different pupillary signature. This position is further strengthened by the analyses we performed using different switch indicators (key presses and on-screen switch events; Appendix 1-figure 5), which show data patterns that are fully consistent with those based on the switch-related algorithm.

3) Graphical presentation of dataAll reviewers had a hard time following some of the figures, especially Figure 2. Below is a list of points to address, some specific to Figure 2, others more general. Perhaps most importantly: figures in this manuscript do not convey information clearly. Showing data from an example/representative subject and then distributions (or boxplots or scatter plots) of the relevant parameters could bring out the main effects in a visually convincing way. It seems a bit unusual to show only population averaged time-series as result plots (as all main figures are in this manuscript). Please consider this suggestion carefully.1. Figure 2 is too dense. The reviewers suggest to split this figure (and figure 3) into separate figures for pupil area and change so the reader can directly compare graphs.

We are sorry about the issues with our figures, and we have now made some substantial revisions to them. We have followed recommendation #1 and have split Figures 2 and 3 into separate panels devoted to pupil area and to pupil change rate, respectively. We have also reduced clutter by removing the insets from Figure 2 that showed pupil responses time locked to key press events, and moved those to a separate figure that also includes new plots that show pupil responses time locked to on-screen switches (those latter plots were added in response to reviewer point 3.3 below).

2. Additionally, there are some issues with the labeling in Figure 2. Minimally, it would be helpful to label constriction and de-dilation episodes that are referred to in the text for all four conditions so text in lines 190-192 can be matched with plots: "The fact that only the Report conditions show the re-dilation going past baseline and being followed by a final constriction".

That is an excellent idea. We have now labeled specific pupillary events in Figure 2 so that we can refer to those labels in the main text (pages 5-8), as well as in the figure caption. In addition, across all relevant figures we have now changed the formatting so that each 1-second interval is easily distinguished from the next (by background shading), both in the panels that show pupil traces, and in the separate timelines that show between-condition statistics. This makes it easier to mentally align various plots and panels with each other and thereby verify temporal overlap (or lack of it) between data patterns shown in separate plots and panels.

3. Figure 2. It would be also useful to have same plots centered at the moment of an actual change for on-screen condition.

We like this idea, and we have now implemented it. As briefly mentioned above we have combined these 'on-screen switch-centered' plots with the 'key-centered' plots within a single figure that also shows the 'OKN-centered' plots so that the results of the three methods can be easily compared (Appendix 1-figure 5).

4. Figure 2. There is plenty of space for the central figure, so no point in abbreviating condition into "Ign.", "On-sc.", etc.

Agreed. This has now been changed.

5. For the average traces of figure 2, how do the authors deal with the fact that a single datapoint can contribute at multiple time points, and this effect is stronger for fast switchers than for slow switchers? Some of the previous studies have cut the data half-way to the next switch to avoid this issue, although this is not completely free of issues, either. If no such step is done, this could shift the mean of the average curve relative to 0 (the overall mean is of course zero due to z-scoring, more data points close to switches contribute more likely multiple times), which does not seem to be the case here (but I do not see a switch-wise baseline-correction either, which would in itself be problematic, as then the constriction could be in fact a rebound from a previous dilation). Maybe the authors could add some lines on the averaging procedure, just to make sure that these issues are not missed. I understand that the section "Artefacts from overlapping pupil responses" addresses a similar issue for the later analyses, but a direct comment on this for figure 2 would be helpful (and maybe a replotting of figure 2 with the half-way cut in the Appendix?).

We have now modified the text to provide additional discussion of this issue (page 4-5, right below Figure 1), i.e. of the possibility that events may contribute at multiple time points within the examined time window. When it comes to this issue, a key difference between our study and some earlier studies is that some earlier studies averaged pupil traces across time windows time-locked to switch events, whereas we performed a general linear model analysis to estimate the pupil time course associated with switch events. Although the two approaches have essentially the same objective, the critical benefit of a general linear model approach in this context is that it is specifically equipped to disentangle overlapping responses linked to closely spaced events. In a nutshell, general linear model approaches (in this case) estimate the event-linked pupil response that best explains the observed pupil time course while, importantly, assuming linear addition of overlapping responses. In other words, whereas the peri-event averaging approach that has been used in previous studies will cause events to contribute to the average trace multiple times if events lie sufficiently close to each other to fall within the peri-event time window under study, this is not true for a general linear model approach: a general linear model approach estimates which event-linked pupil response fits the observed pupil time series best while assuming that overlapping responses (linked to multiple, closely-spaced events) will sum to produce the observed time series.

6. Figure 2. The amplitude and time-courses of the positive rate of change of pupil following constriction appear to be comparable in the Ignore and Report conditions. I might be missing something here, but I'm not clear how this is consistent with different pupil dilations around these epochs (i.e. when the pupil rate of change is positive). That is: the same positive rate of change over the same period of time should lead to the same amplitude of dilation.

We agree: there should clearly be a direct correspondence between the pupil area curves and the derivative-of-area curves. When comparing the sections of the curves identified in this review comment (i.e. sections of positive area change following the initial constriction) we, indeed, see similar levels for the rate curves across Ignore and Report conditions, indicating similar rates of area change. In the same time window we also see similar slopes for the area curves, as expected. We agree that, in that same time period, the net level (as opposed to slope) of dilation that is visible in the area curves is larger for the Report conditions (for which the curves indicate a relatively large pupil in this period) than it is for the Ignore conditions (for which the curves indicate a relatively small pupil in this period). We believe that the reason for this is not that the pupil dilates much more extensively during this period for the Report conditions (instead, the slopes of the area curves are similar), but that the pupil starts its area increase during this period from a considerably less constricted state in those conditions. Reading off the plots, the smallest pupil size immediately before the start of the dilation period in question lies between 0.1 and 0.2 standard deviations below the average in the Ignore conditions, versus fewer than 0.1 standard deviations below the average for the Report conditions. This explains why, following a subsequent period of positive size change that is fairly similar across all conditions, the pupil ends up being more dilated in the Report conditions.

We think that act of examining correspondence between the area curves and the change rate curves (as done by the reviewer and probably by future readers) should be made easier by one of the above-mentioned formatting changes we now made, namely the change that helps identify corresponding 1-second time periods within separate plots.

Tying these observations back to our overall conclusions: even though these considerations highlight that switches are followed by a more extensive net constriction in the Ignore conditions, we should not conclude that the Ignore and Report conditions differ from each other in terms the switch-related constriction response; rather, in the Report conditions this switch-related constriction response overlaps with a response-related dilation response, leading to a smaller net constriction.

7. Figure 2 (and following figures). In addition to the pupil size and rate of change, did the authors try and use the time to peak or maximum rate of change (dilation or constriction) as a measure to compare between conditions?

When we wrote our original manuscript we did not try these measures. Figure 2 does show, to some extent, that the maximum rate of change statistically differs between conditions in various cases, in the sense that, in those cases, change rate reaches its maximum at a time point that falls within the time period where change rate significantly differs between conditions. We have now computed the actual values of the maximum and minimum rates of change in the four conditions, and added a discussion of those values and their interpretation to the caption of Figure 2.

8. Figure 2. For the epoch preceding a switch, the form of the pupil- and rate of change of pupil-timeseries appear quite different to me, in both the Rivalry conditions. That is, in the 2 panels in the left column of figure 2, the Report condition seems to be associated with a sustained dilation before and beyond t=0 while in the Ignore condition, the pupil starts constricting before t=0. Is this difference really not reliable (significant in some way) across subjects?

We entirely agree with this assessment. In fact, in an earlier draft version of the manuscript we put more emphasis on the apparent dilation preceding t=0 in the Rivalry conditions. In the end we did not remark on that data pattern in the submitted manuscript because it does not reach significance in our data. In particular, the within-condition p values fall just short of significance, and the between-condition Rivalry vs On-screen comparisons fall well short of significance. We do agree, though, that it's an interesting pattern that one might have expected a priori (e.g. perhaps rivalry switches involve a type of 'effort'). In our revised manuscript we now added a separate Discussion paragraph dedicated to the apparent data pattern (pages 18-19), but we do not mention it in the Results section because it remains statistically non-significant.

4) Result clarificationThe reviewers raised a number of specific concerns about the interpretation of results. Some of these issues (those related to the figures) are probably easily resolvable, others might need more clarification and rewriting of parts of the manuscript.1. Page 6, Line 198. "This pattern of…". Could the authors clarify how this claim is supported? The dilation is significant when the pupil data are aligned with the switch and with the response, ie – with the perceptual effect and the response effort. That is, the significant dilation could be associated with either or both: perceptual effect and response effort. I don't see how the result shown here alone can support this claim. I agree with the result supporting the claim that dilation but not constriction is related with response effort, but that isn't too surprising since the former precedes the latter.

We agree with the reviewer that, when it comes to the idea that the constriction is switch-related and the dilation is response-related, the analysis of response-locked pupil signals does not add a lot of further evidence beyond what can already be gleaned from the signals that are locked to the OKN-based switch moments. That consideration, combined with the reviewer recommendations (above) to also show data time-locked to the on-screen switches and to clean up Figure 2, has motivated us to stick to only OKN-based switches in the main text, and to combine the response-locked and on-screen switch-locked data together in an Appendix figure (Appendix 1-Figure 5). As part of that change, the text that the reviewer is commenting on in this reviewer point has also been removed. (For completeness we'll add: the review report correctly points out that the dilation is significant regardless of whether the OKN-based time reference or the report-based time reference is used, but this is not inconsistent with the interpretation that the dilation is causally linked to the report, because the two time references are temporally correlated.)

2. Figure 3. The paired t-test shows a reliable difference between the rivalry and on-screen conditions at the end of the constriction phase and through the early part of the dilation phase (as early as about 1 sec). To me, this suggests a significant contribution of dilation related neural activity when comparing rivalrous and explicit changes in the stimulus. Isn't this contrary to the main claim that pupil-linked arousal (presumably involving noradrenergic mechanisms, among others) is not related to spontaneous switching? Alternatively, if I'm misreading the numbers and position of the black line, so that it does not cover the dilation phase, this is another reason to show the data another way. For example, plot the distribution of pupil rate of change for the significant epoch so that one can decipher whether it is a dilation or constriction phase.

Correct, that paired t-test does show a reliable difference between the rivalry and on-screen conditions in that time period. The difference reflects the fact that both the rate of constriction and the rate of dilation immediately following the switch are more extreme in the On-screen conditions than in the Rivalry conditions. We now explicitly address this difference in the caption of Figure 2, and also in a paragraph immediately below that figure (page 8). We would not want to make too much of this qualitative difference, because it seems quite possible to us that the On-screen switches constitute a more crisp and abrupt event than the Rivalry switches, which would result in a more 'smeared out' pupil response estimate in the Rivalry conditions. We now mention this in that nearby paragraph. As for the issue of potentially misreading numbers in the way we presented the data previously: we hope that the new addition of the one-second interval markers (using background shading, mentioned above) makes it easier to discern which significant effects and curve sections are temporally aligned.

3. Figure 4B. It's likely that this is due to using the mean+-sem to display the data, but when I look at the top and bottom panels of this figure, it appears that there is a large dilation on the bottom left compared with top left (Report and Ignore for Rivalry switches respectively) and on the top right compared with bottom right (Ignore and Report for On-Screen switches, respectively). Since these are shown as not-significant from the ANOVA analysis, can the authors reconcile why there appears to be a relatively large, nonzero mean+-sem range for dilation (at about 2-4 sec)?

Yes, we agree that the qualitative patterns are there in the plots as described in the reviewer point. The observation (in the reviewer point) that the dilation is larger in the bottom left plot than in the top left plot, but larger in the top right plot than in the bottom right plot, would translate to an interaction in the ANOVA. The observation (in the reviewer point) that there is a section where mean+-sem is positive in individual curves would translate to a t-test result for those individual curves. We have checked the outcomes of the relevant tests, and they do not show any significant effects in the relevant time period. To be more specific, for the ANOVA analysis no single dilation period was associated with an interaction effect with a cluster-based p value smaller than 0.20 (the p value below which our analysis code outputs the exact p value; our α level was much lower at 0.01). For the per-curve t-tests, on the other hand, the time period indicated in the reviewer point is associated with relatively low p values for both of the curves highlighted in the reviewer point: for Rivalry Report (bottom left) the p value is 0.18, and for On-screen Ignore (top right) the p value is 0.13. Neither is below our chosen α level of 0.01.

With regard to the relation between the error bars in the plots and the p values of the statistical tests: as the review report correctly points out, the error bars are standard errors of the sample mean per timepoint. For inferential statistics, on the other hand, we used cluster-based p values (see main text page pages 25-26 for details) to avoid the multiple comparison issue associated with per-timepoint statistics for data like these. This makes it non-straightforward to estimate from the plots' error bars whether a given section of a curve will be associated with a significant effect in our tests. For the ANOVA such estimates are further complicated by the fact that these are repeated-measures ANOVAs (i.e. maintaining participant identity when comparing between conditions) whereas the error bars indicate across-participant spread within conditions (i.e. spread that is orthogonal to the within-participant, across-condition spread that is relevant to the ANOVA), and also by the fact that the ANOVA concerns comparisons between conditions, so that the exact temporal alignment of apparent data patterns across curves becomes important.

4. Figure 4B. Similar to point 4 above, the bar indicating significance appears to bridge the inflexion point for the pupil derivative – suggesting that the late constriction and early dilation phases, following the switch, are reliably related to the switch. Again – this is contrary to the main claim that arousal (or noradrenaline) linked dilation is not associated with the switch. Could the authors clarify this?

One part of our response here is related to the last part of our response to reviewer point 3.6. We would argue that the question of whether the significant time period here straddles between the late net-constriction phase and the early net-dilation phase cannot in itself be used to distinguish whether the significant effect is on the putative switch-related constriction component or on the putative response-related dilation, because the two putative components overlap in this analysis. A less pronounced, say, switch-related constriction response could translate to a slower net constriction as well as to a more rapid net dilation. A second part of our response is that the particular analysis illustrated in figure 4B concerns the relation between the duration of the preceding dominance period and the pupil change rate, so we would argue that the analysis doesn't really speak to whether the switch is associated with arousal-linked pupil dilation or not. This latter consideration makes us wonder whether we are misunderstanding the point being raised in this point, so our apologies if our response does not fully address the concern, and we'll be happy to address it better if needed.

On a side note: the formatting changes we made in response to reviewer point 3.2 now make it much easier to see from Figure 4B whether or not the significant region bridges the inflection point between constriction and dilation to begin with.

5) Pupil pathways / result interpretationThe reviewers would like to encourage the authors to try and include more of what we do know about neuromodulation and the cortical control of pupil pathways to frame the hypothesis and interpret the results.The discussion leading to the conclusion about the relative roles of parasympathetic and sympathetic systems is somewhat weak. Since this is claimed as a main finding, it needs work both to improve the accuracy and strength of the argument. For example, line 409: "… cortical inhibition of the Edinger-Westphal nucleus,.…": there is no direct cortical inhibition of the EWN (or any region). Previous work has shown both: pupil dilation and constriction resulting from cortical (frontal eye fields, FEF) microstimulation (Lehmann and Corneil, 2016; Ebitz and Moore, 2017). One pathway for such effects is FEF projections to the pretectal olivary nucleus that then modulates activity in the EWN. Another pathway for cortical effects to reach the EWN is via the superior colliculus.

We apologize for the cursory treatment, in our original manuscript, of potential pathways involved, and for our inaccurate statement that cortex itself exerts inhibition on the Edinger-Westphal nucleus. We have now corrected and extended discussion on this topic (page 16). To briefly summarize the contents of that section: we discuss potential routes involving the superior colliculus, involving the locus coeruleus, and involving the pretectal olivary nucleus. Previous research has provided evidence against the latter option, based on the observation that pupil constriction in response to isoluminant visual input changes are preserved in patients with damage to the pretectal area. That would leave routes to the Edinger-Westphal nucleus that involve the superior colliculus and/or the locus coeruleus (please also see our response to the next reviewer point).

In the view of one of the reviewers, one convincing argument consistent with their findings (and that would allow them hypothesize about the putative pathways) could center on the idea of transient lapses in attention (or in pupil linked arousal). The Discussion starting on line 387 starts out in this direction. For example, transient reductions in the activity of the noradrenergic system could lead to reduced inhibition of EWN (that is effected via α-2 adrenergic receptors on EWN neurons). So, in addition to any cortical effects (possibly via the pretectal olivary nucleus) on the parasympathetic system, there could reductions in the activation of the sympathetic pathways, leading to disinhibition of the EWN. This would result in pupil constriction consistent with the results shown here.Related to the above point, the Discussion (paragraph starting on line 368) cites previous work that found dilation linked with perceptual switching (or cognitive switching, in some of the references). However, in some of those studies, the dilation linked effects were derived from measures of baseline pupil size. Those studies predicated their analyses on this measure (baseline pupil), using it as an indicator of arousal (or attentional) state preceding action-selection, and they found reliable links between pupil linked brain state and successful detection of changes in stimulus (or stimulus reward associations, eg change-points in explore-exploit choice tasks). Do these data allow for a baseline contingent analysis?

The reviewer is absolutely right. In fact, an explanation in terms of transient reductions in the noradrenergic system's inhibition of the Edinger-Westphal nucleus is the option that seems to have the most support among authors of existing work on constriction in response to isoluminant visual input changes (to our knowledge the idea is most explicitly articulated in Wilhelm et al. (2002) Brain 125: 2296-2307). So, the full hypothesis would then be that when visual cortex changes its response (be it due to an input-side change or in association with a binocular rivalry switch), this leads to an interruption of steady-state sympathetic inhibition exerted on the Edinger-Westphal nucleus, plausibly involving the locus coeruleus, and this then results in transient pupil constriction. In addition to this option involving the locus coeruleus, we believe that a route involving the superior colliculus cannot be ruled out. We apologize that our original discussion did not convey these considerations well. We have now adjusted our Discussion section (page 16) to convey it better.

We note that, while our original framing of the results contrasted existing interpretations of switch-related pupil responses that are in terms of the noradrenergic locus coeruleus system, with our finding of switch-related constriction, we now nuance that contrast in light of the updated discussion. In particular, while the observed constriction is clearly not consistent with a surge in noradrenaline (which would cause dilation), we now highlight the possibility that this constriction, itself, might also involve the sympathetic system centered on the locus coeruleus, as it might reflect a transient downmodulation of the activity of that system (updated manuscript, page 16).

In the context of the present study we have not been able to conduct an easily interpretable analysis contingent on baseline pupil measures, as suggested in the reviewer point, because the experimental design was not optimized for that purpose. For instance, the necessarily close and inconsistent spacing in our design of separate events that each may affect pupil size renders it difficult to extract a baseline size associated with a given event. Another complication for such an analysis in the context of this particular project is that we have no reliable estimate of absolute pupil size because we did not use an artificial pupil and the distance between tracker camera and pupil could vary somewhat between trials and sessions (all our analyses are in terms of within-trial z-scores). But we fully agree with the reviewer point that an assessment of baseline dependence is likely to be informative in future work.

[Editors' note: further revisions were suggested prior to acceptance, as described below.]

Essential revisions:1) It is critical that the authors please revisit their statistical tests, and how they are reported. It is still unclear whether the reported statistics are calculated within or across subjects, and whether tests are done on combined data.

We apologize for the lack of clarity here. We have now changed the text to convey more effectively what statistics were done, in the Methods section labeled 'Statistics' (manuscript page 27). Here follows a summary of what that updated text describes. First all event-related pupil time courses were computed within participants, after which all statistical tests were performed as fixed-effects tests on the across-participant sets of values (i.e. on the across-participant empirical distributions of values). Performing these tests (t-tests and ANOVAs) for each individual time point within an event-related pupil signal would result in a multiple comparisons problem, which is why we computed cluster-level p-values. In essence, this means that the reported p-values indicate the probability under the null, not of observing an across-participant set of data points that is as extreme as the observed set of data points, but of observing a cluster, contiguous across time, of extreme across-participant sets of data points that is as large as the observed cluster. The extended manuscript text describes this procedure in more detail (manuscript page 27).

If yes, the authors need to also report within-subject effects. Rev. 2 and the reviewing editor still have some doubt about the reliability of effects, and the authors need to pay close attention to this issue before resubmitting their revision.

We thank the Editor for this recommendation. We also appreciate the Editor's related recommendation in the review report of the original submission (not the current one), which read: "Showing data from an example/representative subject and then distributions (or boxplots or scatter plots) of the relevant parameters could bring out the main effects in a visually convincing way." We apologize for our oversight of not adequately incorporating that previous recommendation in our first revision. In response to these recommendations, we have now made two substantive changes to the manuscript. First, we have added a new figure and panel (current Figure 3 and current Figure 4C) that summarize how each individual participant scores on each of the effects that is reported as significant based on across-participant one-sample t-tests. In other words: these new graphics show, for each participant individually, what the average level of the pupil response curve is within each time window during which the pupil response curve, assessed across all participants, significantly differs from 0. Note, 'pupil response curve' here refers to the curve that displays the first temporal derivative of pupil size, which is the curve associated with all statistical tests in our study. Aside from each participant's individual value for a given effect, the new graphics also display the across-participant median of those values. This makes it easier to see, for each time interval during which a pupil response curve, assessed across all participants, significantly lies above or below 0, that the response curves of individual participants also lie on the same side of 0 for over half of the participants (i.e. the median consistently lies on the expected side of 0). Second, aside from adding these graphics to the main text, and editing the main text accordingly (manuscript pages 8 and 11), we have also added a new figure to the Appendix (Appendix Figure 6). This figure shows the same data as shown in Figure 2A of the main text, but instead of showing the across-participant average curves (as Figure 2A does), the new Appendix figure shows the pupil response curves of all individual participants that make up this average. As described in the newly added Appendix text that accompanies this figure (Appendix pages 9 and 10), these within-participant curves reinforce the notion that the most important data patterns reported in the main text are present in the majority of individual observers. The within-participant curves also convey additional information about between-observer differences that was not conveyed by the across-observer averages we used so far (discussed on Appendix pages 9 and 10).

2) Clarify and add to the description of OKN analysis, as described in detail by rev. 3 below.

We thank the Editor for this suggestion. We have edited the main text (bottom of manuscript page 20) based on the text that was in our original reply to Reviewers, following the recommendation by Reviewer #3.

Reviewer #2:It is still unclear to me (after reading the Methods/Data Analysis section) whether the reported statistics are calculated within or across subjects. I appreciate the responses to questions in the initial review, but I think this needs more clarity. For example, the responses in "Result clarification": how are these tests done?

We apologize for the lack of clarity. All reported statistical tests are across-participant tests. This is now mentioned explicitly in the Methods section, along with other newly added clarifications regarding the statistical tests (manuscript page 27).

It looks like they are done on the combined data. But are there within subject effects and if so, what proportion of subjects show the main effects that are highlighted in the Results section? This is an important point because if the Conclusion about the general applicability of these findings is to be, the findings needs to be reliable in at least a majority of subjects.

Correct, the tests were performed on the across-participant distributions of data values. In response to this request, and also in response to the related point in the Editor's remarks (Essential Revision 1), we have now added a figure (the new Figure 3) and a panel (the new panel 4C) to the main text. These added graphics depict how the individual participants score on each of the effects that are reported as significant on the basis of across-participant one-sample t-tests. In other words: for each participant individually, the graphics depict the average value of the pupil change rate curve within every time window for which the across-participant statistics indicate a pupil change rate significantly different from 0. These newly added graphics show that, for each of these time windows, a majority of participants have a pupil change rate that lies on the same side of 0 as the across-participant average does. In other words: whenever a dilation (or constriction) is marked as significant on the basis of the across-participant statistics, a majority of participants individually show a dilation (or constriction) in that time window. To make this particularly clear in the added figures, these show not only the within-participant values but also the medians of these values -- these medians consistently lie on the expected side of 0. We also inserted additional text in the Results section, describing what the newly added graphics show (manuscript pages 8 and 11). Finally, in the Appendix we now show all within-participant event related pupil time courses that underlie main text Figure 2A (Appendix Figure 6). The figures show that the qualitative data patterns highlighted in association with Figure 2A are also present in a majority of individual participants, as now discussed in the Appendix text surrounding the new Appendix figure (Appendix pages 9 and 10). That same Appendix text also describes additional interesting aspects of between-observer variability in the observed pupil responses; aspects that were not conveyed by our across-observer averages.

The expanded Discussion is an improvement. Another point that might be helpful to consider is that it is mostly the prefrontal and cingulate cortices that provide descending control of LC excitability. Since this study is centered on perceptual effects and not on cognitive or physical effort, the scope of the work is consistent with a sensory cortical dominant network so that cortical regulation of LC-linked arousal is not a factor.

We thank the reviewer for this recommendation. We have made a further addition to our Discussion section to point out the discrepancy, when it comes to a LC-centered explanation of the switch-related constriction, between the dominant origin of cortical inputs to the LC (frontal and cingulate cortices) and the plausible cortical regions involved in the visual events that trigger this constriction (posterior regions). This added text is on manuscript page 17.

Line 178: Rate of change was used in an earlier study to show LC spike linked changes in pupil size in Joshi et al., 2016 (their Figures 4-6 and 8).

Thank you for this pointer. A reference to the paper in question has now been inserted at the relevant location in the manuscript (page 5).

Reviewer #3:The authors address all my concerns. The only change I would suggest is using part of their response in the manuscript itself, as it conveys the idea very nicely. I would use it to replace / complement the second half of the last sentence in "Limitations of the OKN-based method of identifying switches". The part of the response is "Finally, although the OKN-based algorithm, then, is not perfect, for our purposes the critical requirement is that the OKN-based algorithm marks time points that are strongly associated with perceptual switches; not that it marks zero spurious time points nor that it marks a time point for every single perceptual switch. This is especially true because our key analyses concern comparisons between conditions while keeping the switch-identification method constant; not comparisons between identification methods."

We thank the Reviewer for this suggestion. We have edited the main text (bottom of manuscript page 20) based on the text that was in our original reply to Reviewers, following this recommendation.